# A chromosome-level genome assembly for the amphibious plant *Rorippa aquatica* reveals its allotetraploid origin and mechanisms of heterophylly upon submergence

Tomoaki Sakamoto [1,2], Shuka Ikematsu [1,2], Hokuto Nakayama [1,3,4], Terezie Mandáková[5], Gholamreza Gohari[6], Takuya Sakamoto [7,10], Gaojie Li[8], Hongwei Hou [8], Sachihiro Matsunaga [9], Martin A. Lysak [5] & Seisuke Kimura [1,2] ✉

The ability to respond to varying environments is crucial for sessile organisms such as plants. The amphibious plant *Rorippa aquatica* exhibits a striking type of phenotypic plasticity known as heterophylly, a phenomenon in which leaf form is altered in response to environmental factors. However, the underlying molecular mechanisms of heterophylly are yet to be fully understood. To uncover the genetic basis and analyze the evolutionary processes driving heterophylly in *R. aquatica*, we assembled the chromosome-level genome of the species. Comparative chromosome painting and chromosomal genomics revealed that allopolyploidization and subsequent post-polyploid descending dysploidy occurred during the speciation of *R. aquatica*. Based on the obtained genomic data, the transcriptome analyses revealed that ethylene signaling plays a central role in regulating heterophylly under submerged conditions, with blue light signaling acting as an attenuator of ethylene signal. The assembled *R. aquatica* reference genome provides insights into the molecular mechanisms and evolution of heterophylly.

Land plants evolved from aquatic green algae approximately 450 million years ago. Some land plants re-entered and adapted to aquatic environments and are called aquatic plants. Aquatic plants, also known as hydrophytes or macrophytes, evolved various traits to survive in aquatic habitats, such as narrow and flexible leaf blades for efficient gas exchange, aerenchyma formation, and a high capacity for vegetative reproduction.

Among aquatic plants, there exists a group referred to as amphibious plants. These plants are primarily found in environments characterized by fluctuating water levels, such as at the edge of water, and are capable of thriving both in terrestrial and submerged conditions. To adapt to these fluctuating environments, phenotypic plasticity plays a crucial role in amphibious plants. This is particularly evident in the phenomenon called heterophylly, in which a marked change occurs in leaf shape and function between terrestrial and submerged conditions[1,2]. Submerged leaves are often more finely dissected and narrower than terrestrial leaves[3,4], and stomatal formation is suppressed underwater[5–7], allowing for efficient photosynthesis

[1]Faculty of Life Sciences, Kyoto Sangyo University, Kamigamo-Motoyama, Kita-ku, Kyoto, Japan. [2]Center for Plant Sciences, Kyoto Sangyo University, Kamigamo-Motoyama, Kita-ku, Kyoto, Japan. [3]Graduate School of Science, Department of Biological Sciences, The University of Tokyo, Science Build. #2, 7-3-1 Hongo, Bunkyo-ku, Tokyo, Japan. [4]Department of Plant Biology, University of California Davis, One Shields Avenue, Davis, CA, USA. [5]CEITEC – Central European Institute of Technology, Masaryk University, CZ-625 00, Brno, Czech Republic. [6]Department of Horticulture, Faculty of Agriculture, University of Maragheh, Maragheh, Iran. [7]Department of Applied Biological Science, Faculty of Science and Technology, Tokyo University of Science, 2641 Yamazaki, Noda, Chiba, Japan. [8]The Key Laboratory of Aquatic Biodiversity and Conservation of Chinese Academy of Sciences, Institute of Hydrobiology, Chinese Academy of Sciences, Wuhan, Hubei, China. [9]Department of Integrated Biosciences, Graduate School of Frontier Science, The University of Tokyo, Chiba, Japan. [10]Present address: Faculty of Science, Kanagawa University, 3-27-1 Rokkakubashi, Kanagawa-ku, Yokohama, Kanagawa, Japan. ✉e-mail: seisuke@cc.kyoto-su.ac.jp

in both aquatic and terrestrial environments. Studying amphibious plants can provide valuable insight into the mechanisms of adaptation and evolutionary processes of plants in aquatic environments.

The adaptive traits of amphibious plants have long been intriguing botanists[1,8–11]. Several new models species, such as *Rorippa aquatica* (tribe Cardamineae, Brassicaceae)[3], *Hygrophila difformis* (Acanthaceae)[4,12], *Ranunculus trichophyllus* (Ranunculaceae)[6], and *Callitriche palustris* (Callitricheae)[13] were developed for studying amphibious plant adaptations. However, despite extensive physiological and morphological studies on this topic[1,8–11], the molecular mechanisms remain largely enigmatic, possibly due to a lack of genomic information for these model amphibious plants.

*Rorippa aquatica*, an amphibious plant found in North American bays of lakes, ponds, and streams[14], exhibits dramatic heterophylly in response to various environmental cues, such as temperature, light quantity, and submergence[3]. The leaves of this species become more deeply dissected and narrower underwater than in the air (Fig. 1a, b). An earlier study demonstrated the mechanism of heterophylly in response to temperature in *R. aquatica*. Changes in the expression of the *KNOTTED1-LIKE HOMEOBOX* (*KNOX1*) gene in response to temperature and light intensity led to altered concentrations of gibberellins and cytokinin in leaf primordia, which in turn

alters leaf morphology[3]. We have previously reported that the somatic cells of *R. aquatica* have 30 chromosomes[15], whereas the base chromosome number (x) in related Cardamineae species is eight and diploid species have $2n = 2x = 16$[16], suggesting that *R. aquatica* has a polyploid origin. Furthermore, the fact that the chromosome number of *R. aquatica* is not a multiple of eight indicates that its genome was restructured after polyploidization.

In addition to exhibiting remarkable heterophylly, *R. aquatica* propagates vegetatively in nature. It produces flowers with white petals from April to August in its natural habitat[17], and inducing floral differentiation under low-temperature conditions can be achieved in laboratory settings[18]. The rarity of seed production may be caused by self-incompatibility[17], which leads to the formation of self-clonal colonies. This species can regenerate from detached leaf fragments and is a subject of research in the field of plant regeneration[19]. Furthermore, *R. aquatica* is a member of the Brassicaceae and is closely related to *Arabidopsis thaliana* and *Cardamine hirsuta* (tribe Cardamineae, Brassicaceae), a model plant for compound leaf development[20–22], making it suitable as an experimental material. Therefore, if genomic information on *R. aquatica* is revealed, it will not only contribute to the elucidation of the molecular mechanisms and evolutionary basis of adaptation of amphibious plants, but also to breeding studies on improving flood tolerance in Brassicaceae crops (cabbage, turnip, rapeseed, etc.).

In the present study, we assembled the genome of *R. aquatica* at the chromosome level, revealed its chromosome architecture, and reconstructed its origin and evolution. This is the first study in which genomic information was completed at the chromosome level for an amphibious plant that exhibits remarkable heterophylly. In addition, we performed the transcriptome analysis using genomic information to reveal the mechanisms of heterophylly in response to submergence. Our results may shed light on the molecular basis of amphibious plant adaptations to fluctuating environments.

## Results

### Chromosome architecture of *Rorippa aquatica* revealed via comparative cytogenomics

*R. aquatica* has 15 chromosome pairs ($2n = 30$, hereafter listed as chromosomes RaChr01 to RaChr15)[15] (Fig. 1c). We examined the *R. aquatica* genome structure and evolutionary processes via comparative chromosome painting (CCP) based on the localization of contigs of chromosome-specific Bacterial Artificial Chromosome (BAC) clones of *A. thaliana* on meiotic (pachytene) chromosomes (see Fig. 1c for examples of CCP). The painting probes were designed to reflect the system of 22 ancestral genomic blocks (GBs, labeled as A to X)[23,24] and eight chromosomes of the ancestral Cardamineae genome[25]. As all 22 GBs were found to be duplicated within the *R. aquatica* haploid chromosome complement, the species has a tetraploid origin (Fig. 1c).

We used CCP to reconstruct a complete comparative cytogenetic map of *R. aquatica* (Fig. 1c) and compared it to that of the ancestral genome of the tribe Cardamineae with eight chromosomes[25]. Fourteen of the 15 chromosome pairs in *R. aquatica* (RaChr01–RaChr14) are shared with the ancestral Cardamineae genome. Due to its polyploid origin, the *R. aquatica* genome contains six pairs of Cardamineae homeologues: AK1 (GBs A + B + C; RaChr01 and RaChr02), AK3 (F + G + H; RaChr04 and RaChr05), AK4 (I + J; RaChr06 and RaChr07), AK5 [(K–L) + (M–N); RaChr08 and RaChr09], AK6/8 (V+Wa+Q + R; RaChr10 and RaChr11), and AK7 (S + T + U; RaChr12 and RaChr13). Chromosomes RaChr03 and RaChr14 are homeologous to ancestral chromosomes AK2 (D + E) and AK8/6 (O + P+Wb+X), respectively. Chromosome RaChr15 (O + P + E + D+Wb+X) originated via nested chromosome insertion (NCI) of the AK2 homeologue into the centromere of the AK8/6 homeologue. At least three paracentric inversions post-dated the NCI event (Fig. 2). Comparative cytogenomic analysis confirmed the tetraploid origin of *R. aquatica* and allowed us to reconstruct the structure of the RaChr15 fusion chromosome.

### Chromosomal genome assembly and annotation

We assembled the *R. aquatica* genome and predicted its gene structure. As reference, we used *R. aquatica* accession N, for which the heterophylly was

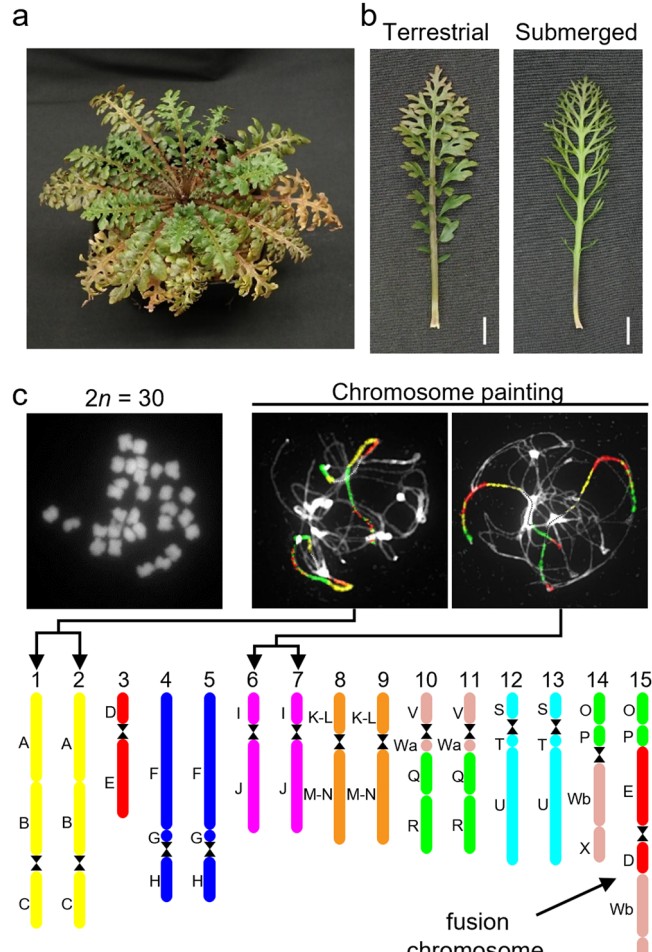

**Fig. 1 | Physiological characteristics and chromosome structure of *R. aquatica*.**
**a** *R. aquatica* grown under terrestrial condition at 25 °C. **b** Expanded leaves of *R. aquatica* grown under terrestrial and submerged conditions at 25 °C (scale bars, 1 cm). **c** Chromosome structure of *R. aquatica*. DAPI-stained mitotic chromosomes prepared from anthers (upper left panel). Chromosome structure (lower panel) was revealed via comparative chromosome painting (Panel **c**, upper right). The different colors in chromosome structure correspond to the ancestral Cardamineae chromosomes, whereas capital letters refer to genomic blocks. See Fig. 2 for detailed structure of the fusion chromosome.

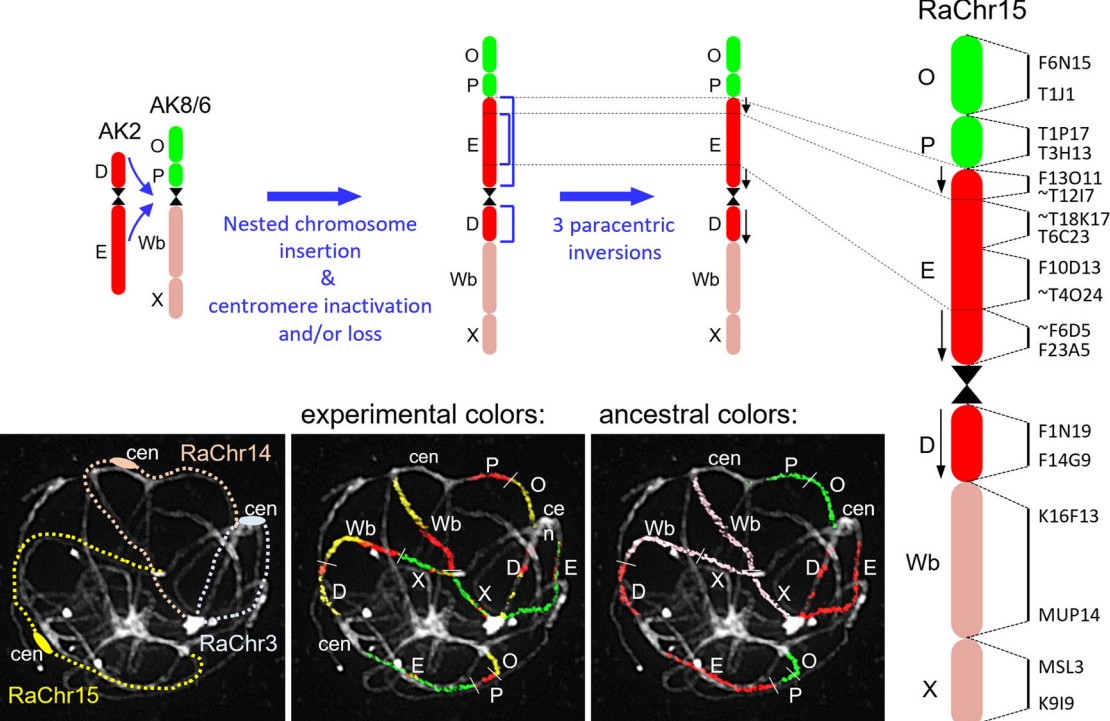

**Fig. 2 | Structure of the fusion chromosome RaChr15 revealed by comparative chromosome painting (CCP) of pachytene chromosomes.** The different colors correspond to the ancestral Cardamineae chromosomes, whereas capital letters refer to genomic blocks. Bacterial Artificial Chromosome (BAC) clones of *Arabidopsis thaliana* defining genomic blocks or their parts are listed along the chromosome

RaChr15. Centromeres are indicated by black hourglass symbols. Blue arrows and staples denote chromosome rearrangements. Black arrows along chromosome indicate the opposite orientation of chromosome regions as compared to that of the ancestral chromosome AK2.

highly responsive to temperature[18]. Hybrid genome assembly with Illumina short reads and PacBio long reads was performed using the MaSuRCA assembler[26]. The assembled draft contigs were scaffolded using Hi-C Seq reads. Hi-C Seq provides information about the physical contact between the genomic loci in the nuclei. Therefore, given that the chromosomes are clustered in the nuclei, Hi-C Seq data scaffolding allows chromosome-level assembly. This generated 15 chromosome-level sequences and 2040 fragments (Supplementary Table 1). The total length of the chromosome sequences and whole sequences were approximately 414 and 452 Mb, respectively, similar to that of the expected genome size (420–450 Mb) obtained using k-mer counting (Supplementary Fig. 1 and Supplementary Table 2). Benchmarking Universal Single-Copy Orthologs (BUSCO) was used to evaluate the assembled genome quality (Supplementary Fig. 2). In total, 1614 conserved single-copy land plant genes were screened within the *R. aquatica* genome, and 99.1% were identified, indicating the high reliability of the assembled genome. Among the 1614 genes, 57.3% of the genes were found to be duplicated. Repeat sequences in the genome were identified (Supplementary Table 3) and masked for subsequent analyses. Gene structures were predicted using the Program to Assemble Spliced Alignments (PASA) pipeline[27,28]. The prediction results obtained using several methods were merged into an integrated gene structure, resulting in the identification of 46,197 genes (Supplementary Table 4). BUSCO assessment to protein sequences from *R. aquatica* gene data identified 94.4% of the conserved genes (Supplementary Fig. 2), indicating the high reliability of the gene annotation. *R. aquatica* has 1.6–1.7 times the number of genes than that of diploid Brassicaceae species such as *A. thaliana* (27,416 genes) and *C. hirsuta* (29,458 genes).

### Evolutionary processes revealed via comparative genomics
Although the ancestral chromosome number of Cardamineae is x = 8, *R. aquatica* has a hypotetraploid chromosome number (2n = 30). This chromosome number and the structure of the *R. aquatica* genome were

elucidated by cytogenomic analyses (Figs. 1c and 2) as a whole-genome duplication followed by a chromosome fusion that reduced the 16 chromosome pairs to 15. To further explore this evolutionary process in *R. aquatica*, a comparative genomics analysis was conducted based on the chromosome sequences.

Using OrthoFinder analysis of the entire genome, a genome-level phylogeny of *R. aquatica* and related species was constructed. The longest protein sequences of each gene were extracted and used as the genome-level protein dataset in the present analysis. The datasets of 24 species were obtained from public genome databases. For intra-genus analyses, the preliminary dataset of *Rorippa islandica* (2n = 16)[29] was prepared by genome assembly, and subsequent gene prediction using public genome-seq read data. Based on the constructed phylogenetic tree, *R. aquatica* and *R. islandica* were placed in the clade that includes *Arabidopsis* (Supplementary Fig. 3), which corresponded to the Brassicaceae clade A[30]. *Barbarea vulgaris* is the closest species, and *C. hirsuta* is the more immediate sister of the genus *Rorippa*, which coincide with past phylogenetic study with some nuclear genes[31].

To investigate the origin of the *R. aquatica* genome, we compared the chromosome-level assemblies of *R. aquatica* and *C. hirsuta* (Supplementary Fig. 4). Multiple alignments based on nucleotide similarity showed that each chromosome of *C. hirsuta* was similar to two *R. aquatica* chromosomes, indicating a tetraploid origin of the latter species. Furthermore, the collinearity of the RaChr15 fusion chromosome with the *C. hirsuta* chromosomes Chr2 and Chr8 indicated that this *R. aquatica* chromosome was formed via an NCI involving the *Rorippa* homeologues RaChr03 and RaChr14, which is consistent with our CCP-based results (Fig. 2).

The divergence ages of duplicated genes could indicate when *R. aquatica* attained its tetraploid-like characteristics. To estimate divergence ages in the Brassicaceae, we selected eight species (*A. thaliana*, *A. lyrata*, *Barbarea vulgaris*, *Capsella rubella*, *Cardamine hirsuta*, *Eutrema salsugineum*, *R. aquatica*, and *R. islandica*). All genes in each genome were

https://doi.org/10.1038/s42003-024-06088-7 **Article**

clustered into orthogroups (i.e., groups consisting of orthologous genes) based on protein similarity (Supplementary Data 1). In total, 10,868 single-copy orthogroups were found by comparing six of the eight species (excluding *R. aquatica* and *R. islandica*). In *R. islandica*, almost all (91.5%) were single-copy genes, whereas in *R. aquatica*, 58.6% were duplicated genes (Supplementary Table 5). This suggests that the *R. aquatica* genome underwent large-scale gene duplication. To estimate species divergence and gene duplication ages, we calculated the synonymous nucleotide substitution rate (Ks) between gene orthologs, both inter- and intra-species. To calculate Ks, we used the longest coding DNA sequences of each gene conserved as a single copy in the seven Brassicaceae species and duplicated in *R. aquatica*; this resulted in 5909 sequences. The Ks distributions of *R. aquatica* compared with those of other species and *R. islandica* were similar, reflecting phylogenetic relatedness (Supplementary Fig. 5). Divergence ages were estimated using Ks as T (in years) = Ks/(2 * $\mu$), where $\mu$ = 6.51648E−09 synonymous substitutions/site/year for Brassicaceae[32]. Based on this result, the divergence times between *Cardamine* and *Rorippa* and between *Barbarea* and *Rorippa* are 13.7–14.2 million years ago (Mya) and 10.5–10.8 Mya, respectively (Supplementary Table 6). The Ks distribution based on the duplicated paralogs of *R. aquatica* has a single peak with a median of Ks = 0.102, which corresponds to a divergence of 7.8 Mya. This suggests that large-scale gene duplication in *R. aquatica* occurred after the *Barbarea/Rorippa* divergence and most likely occurred at the whole-genome level. In *R. aquatica* and *R. islandica*, the median of Ks is 0.073, corresponding to a divergence of 5.6 Mya. Although *R. islandica* has diploid characteristics (Supplementary Table 5) and a chromosome number of 2*n* = 16[29], our findings indicate that these two *Rorippa* species diverged after the whole-genome duplication (WGD).

Ks between *R. aquatica* and *R. islandica* were analyzed at the chromosome level to elucidate this conflict. The chromosomes of *R. aquatica* form two subgenome groups based on their distribution of the calculated Ks relative to that of the *R. islandica* genome (Fig. 3a and Supplementary Tables 7, 8) and they seem to be located in same phylogenetic distances for *B. vulgaris* and *C. hirsuta* due to no clear segregation based on their Ks (Supplementary Fig. 6). The first group, named subgenome A, includes eight chromosomes (RaChr01, -03, -05, -07, -08, -10, -13, and -14) with a median Ks value of approximately 0.05. These chromosomes are closer to those of *R. islandica*. Subgenome B includes seven chromosomes (RaChr02, -04, -06, -09, -11, -12, and -15) with a median Ks value of approximately 0.09. These chromosomes are phylogenetically more distant from those of *R. islandica*; their Ks values relative to *R. islandica* are similar to those between the duplicated *R. aquatica* genes. The fact that *R. aquatica* has two subgenomes with different divergence ages indicates an allotetraploid origin caused by hybridization between two ancestral *Rorippa* species. Integration of various genomic analyses clarified the structure of the *R. aquatica* genome (Fig. 3b). The homologous chromosomes in each subgenome show a similar distribution of genes and long terminal repeats (LTRs). The peaks of the LTR distribution in each chromosome indicate centromeric regions. These results also suggest that gene duplication in the species occurred via a WGD.

Comparative genomics revealed the evolutionary process of establishing the present *R. aquatica* (Fig. 3b): assuming an allotetraploid origin of *R. aquatica*, *Rorippa* split into the subgenome groups A and B approx. 7.2 Mya (Ks = 0.09). In the subgenome A group, divergence into the ancestor of *R. islandica* and the parental species of *R. aquatica* occurred at 4.2 Mya (Ks = 0.05). Subsequent hybridization of two species from different subgenome groups resulted in the formation of the allotetraploid origin of *R. aquatica*. Phylogenetic analysis based on plastid sequences placed *R. islandica* in a clade distant from *R. aquatica*[15]. This suggests a paternal origin of subgenome A of *R. aquatica*. Our analysis does not identify the seed parent of subgenome B. Based on Ks analysis, the fusion chromosome RaChr15 was formed by intra-subgenomic fusion. The median Ks value of RaChr15 versus *R. islandica* was approximately 0.09, which is similar to that of the other chromosomes of subgenome B (Fig. 3a and Supplementary Table 7). Chromosomes RaChr03 and RaChr14, within subgenome A, remained as independent chromosomes.

## Transcriptome analysis reveals the pathway underlying heterophylly in response to submergence

We conducted RNA-seq gene expression analysis using the assembled *R. aquatica* genome data to elucidate the mechanism of heterophylly in response to submergence (information about RNA-seq read data of all transcriptome analyses is shown in Supplementary Data 2). First, we observed the morphology of young leaves over time to determine the timing of leaf shape change upon submergence (Fig. 4a). After one day of submergence, submerged and terrestrial leaves did not differ morphologically. After four days of submergence, the young-leaf margin serrations became deeper in the submerged than in the terrestrial plants. After 7 days of submergence, leaf incisions were significantly deeper in the submerged leaves. To reveal the gene expression patterns of early response to submergence as well as the early stage of leaf morphology differentiation, shoot apices containing young leaves were sampled at 1 h and four days after submergence and RNA-seq analyses were performed. The RNA-seq reads for a 1 h submergence treatment obtained in an earlier study (DRA014113)[33] were incorporated into the analyses. As a result of identification of differentially expressed genes (DEGs) with certain threshold (FDR < 0.01 and |Log$_2$FC| > 1), we identified 787 upregulated and 1091 downregulated genes 1 h after submergence, which increased to 5358 upregulated and 4945 downregulated genes after 4 days of submergence (Fig. 4b). The submergence-responsive DEGs were classified into three classes according to the timing of expression. The genes whose expression changed only within 1 h of submergence were classified as "early response genes." The genes whose expression changed after 1 h as well as after 4 days of submergence were classified as "throughout response genes." The genes whose expression changed only after 4 days of submergence were classified as "late response genes."

Next, we used Gene Ontology (GO) enrichment analysis to elucidate the biological processes involved in regulating heterophylly (Fig. 4b and Supplementary Data 3). Among early response upregulated genes, those in the "shade avoidance" category (GO-ID: 9641) were enriched, suggesting that light conditions are important in the early response to submergence. Significant numbers of both up- and down-regulated genes were related to phytohormones, such as ethylene (GO-ID: 9723), gibberellin (GO-ID: 9739), and abscisic acid (ABA) (GO-ID: 9737). Some down-regulated genes were enriched in "response to auxin stimulus" (GO-ID: 9641). The results are consistent with those of an earlier study[18], which reported that phytohormones regulate heterophylly. Genes related to leaf morphology such as leaf development (GO-ID: 48366) and leaf morphogenesis (GO-ID: 9965) were downregulated in response to submergence. Those involved in adaxial/abaxial axis specification (GO-ID: 9943) were downregulated during the late response. Regulation of cell division and elongation is essential in altering leaf morphology. At the late stage, when the leaf morphology differed between the submerged and terrestrial leaves, genes belonging to the cell cycle (GO-ID: 7049) and cell division (GO-ID: 51301) categories were enriched among the down-regulated genes. The results indicate that the expression of genes involved in leaf development is regulated immediately after submergence and that phytohormones are involved in the regulation.

## Ethylene induces the submerged leaf phenotype

The results of the gene expression analyses suggested that ethylene, gibberellin, and abscisic acid are involved in the regulation of heterophylly. Our earlier studies already showed that gibberellins are involved in the regulation of leaf morphology in *R. aquatica*[3]. Therefore, we examined the effect of ethylene and ABA on leaf shape in *R. aquatica*.

In numerous plant species, the accumulation of ethylene in plant tissues during submergence triggers the submergence response; for instance, ethylene participates in heterophylly in some plant species[5]. Treatment of terrestrial *R. aquatica* plants with 1-aminocyclopropane-1-carboxylic acid (ACC), an ethylene precursor, resulted in the formation of more deeply lobed leaves with narrower leaf blades than that in the untreated terrestrial plants (Fig. 5a). Contrastingly, when the ethylene-response inhibitor

 

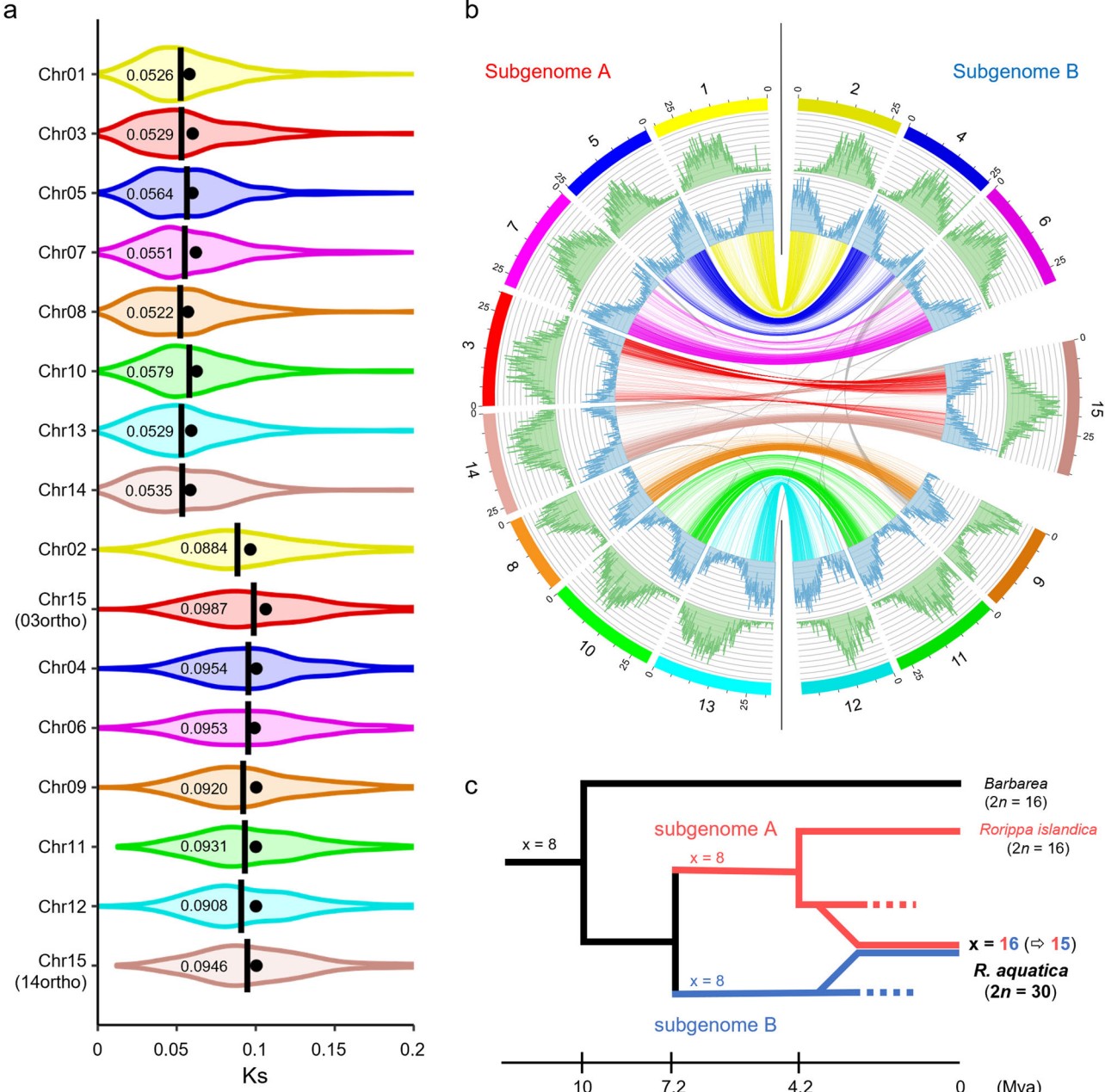

**Fig. 3 | *Rorippa aquatica* chromosome-level genome assembly. a** Chromosome-level synonymous nucleotide substitution rate (Ks) distributions relative to *R. islandica*, for *R. aquatica* paralogs in orthologous chromosomes. Closed circles: mean; bars and numbers: median. **b** Circos plot of the assembled *R. aquatica* genome. In the Circos plot, the long terminal repeat and gene distributions are in green and blue, respectively. The lines at the center link between the paralogous genes for which orthologous genes were conserved as single copy in diploid Brassicaceae species. **c** Evolutionary scheme of the formation the allotetraploid genome of *R. aquatica* based on the current data.

AgNO$_3$ was added under submerged conditions, leaves with expanded leaf blades formed similar to the terrestrial leaves (Fig. 5b). When the ethylene concentration was increased, narrower leaf blades developed (Fig. 5c). The results suggest that heterophylly in *R. aquatica* is regulated by the concentration of the hormone ethylene. Furthermore, the submerged phenotype of leaves was suppressed by the inhibition of ethylene response even under submerged conditions, suggesting that the ethylene response pathway mediates the regulation of heterophylly.

ABA is a well-known hormone involved in the drought response[34] and is also known to be involved in the regulation of heterophylly in several amphibious plant species[4–6,13]. ABA treatment also inhibits *R. aquatica* submergence response in stomatal development[33]. We found that exogenous ABA treatment caused inhibition of leaf shape change in response to submergence (Supplementary Fig. 7). This result indicates that ABA is involved in the regulation of heterophylly in *R. aquatica*, although long-term effects could not be observed because ABA treatment under submerged conditions strongly suppresses plant growth.

Next, we performed RNA-seq analysis of ACC-treated plants to identify the genes responsible for forming submerged type leaves. Because ACC treatment induced the formation of submerged-type leaves, we extracted genes whose expression changed under both submerged and ACC-treated conditions (Fig. 5d, e and Supplementary Data 4): 143 and 82 genes were commonly upregulated and downregulated, respectively. As expected, several common ethylene response genes were upregulated in both datasets. Contrastingly, submergence with AgNO$_3$ showed an inversed expression pattern as correlated to leaf morphology.

**Fig. 4 | Effect of submergence on *Rorippa aquatica* leaf morphology and gene expression. a** Images and outlines of newly emerged young leaves after transfer to terrestrial or submerged conditions (scale bar, 1 cm). **b** Transcriptome analysis of leaves grown under the submerged condition. Differentially Expressed Genes (DEGs) were identified based on significant differences (FDR < 0.01) in expression and log₂(fold change) |Log₂FC| > 1. Based on their expression patterns, DEGs were categorized as "early response genes" (responding only within the first hour), "late-response genes" (after 4 days), and "throughout-response genes" (throughout submergence), then subjected to Gene Ontology enrichment analysis (significantly enriched categories are shown).

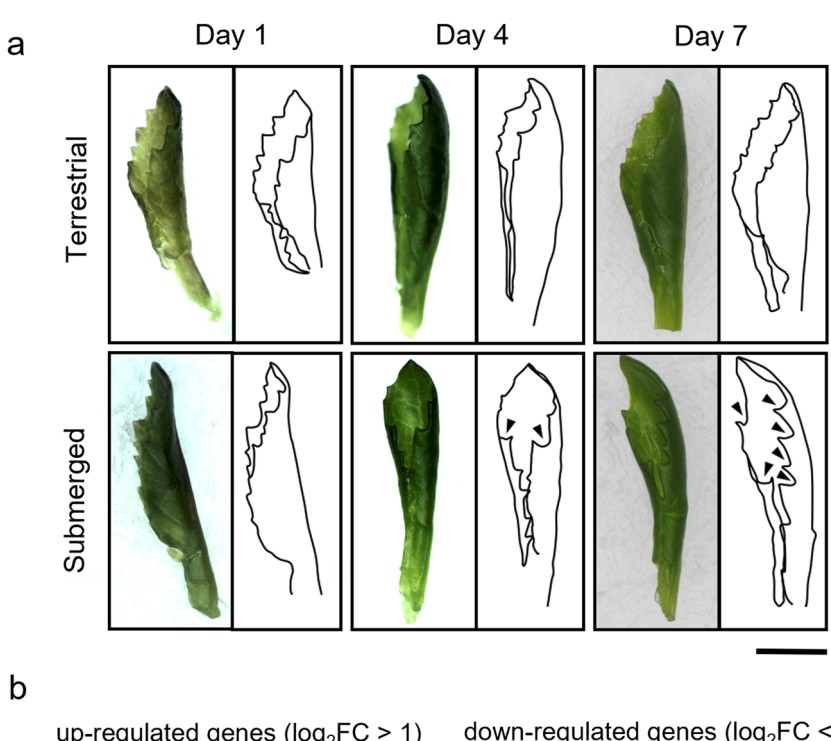

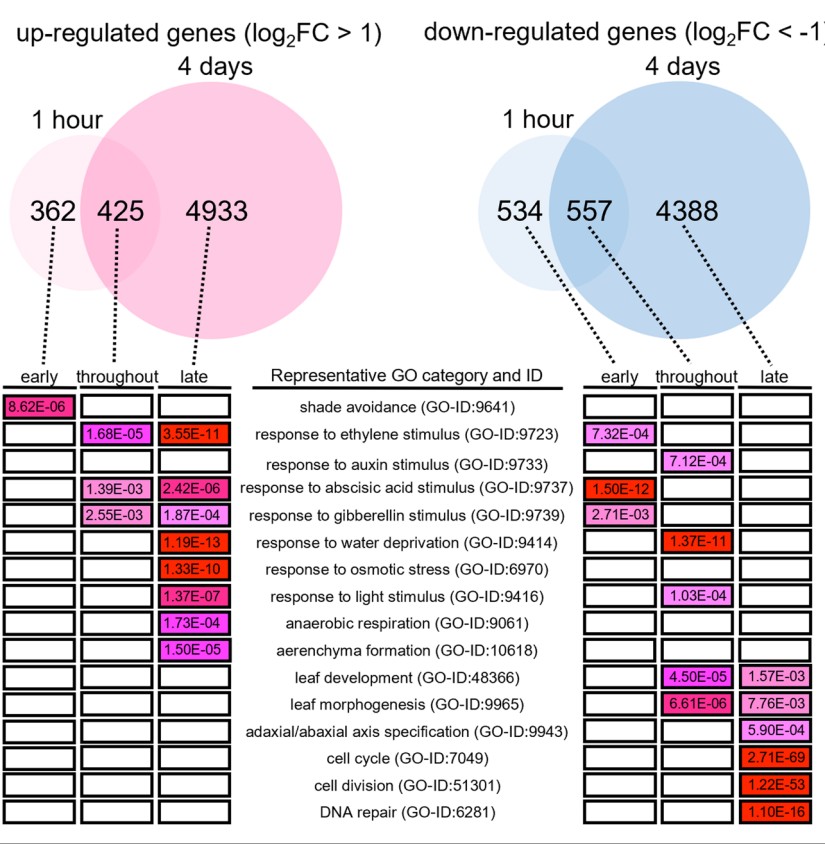

One of the commonly regulated genes, *LONGIFOLIA* (*LNG1* and *LNG2*), was identified to participate in leaf morphogenesis in *A. thaliana* via activation-tag gene screening[35]: The dominant mutant of *LNG1* (*lng1-1D*) formed elongated and narrow leaves with serrated margins; furthermore, *LNG1* and *LNG2* may function redundantly and regulate the longitudinal elongation of cells.

*LATE MERISTEM IDENTITY1* (*LMI1*) and *REDUCED COMPLEXITY* (*RCO*) arose from the same ancestral gene via gene duplication within a clade of Brassicaceae[36]. *RCO* controls leaf complexity in *C. hirsuta*, wherein the wild type has compound leaves, even though the *rco* mutant displays simple lobed leaves[36]. *RCO* is lost and *LMI1* is retained in *A. thaliana*, resulting in simple leaves and the introduction of *C. hirsuta RCO* into *A. thaliana* resulted in the formation of serrations. This *R. aquatica* gene *RaChr03G09000*, extracted as an ortholog of *A. thaliana LMI1*, is an *RCO* ortholog because it has a higher protein identity with *C. hirsuta RCO* (84.3%) than with *C. hirsuta LMI1* (63.1%). Therefore, the upregulation of *R. aquatica RCO* may cause the formation of compound leaves.

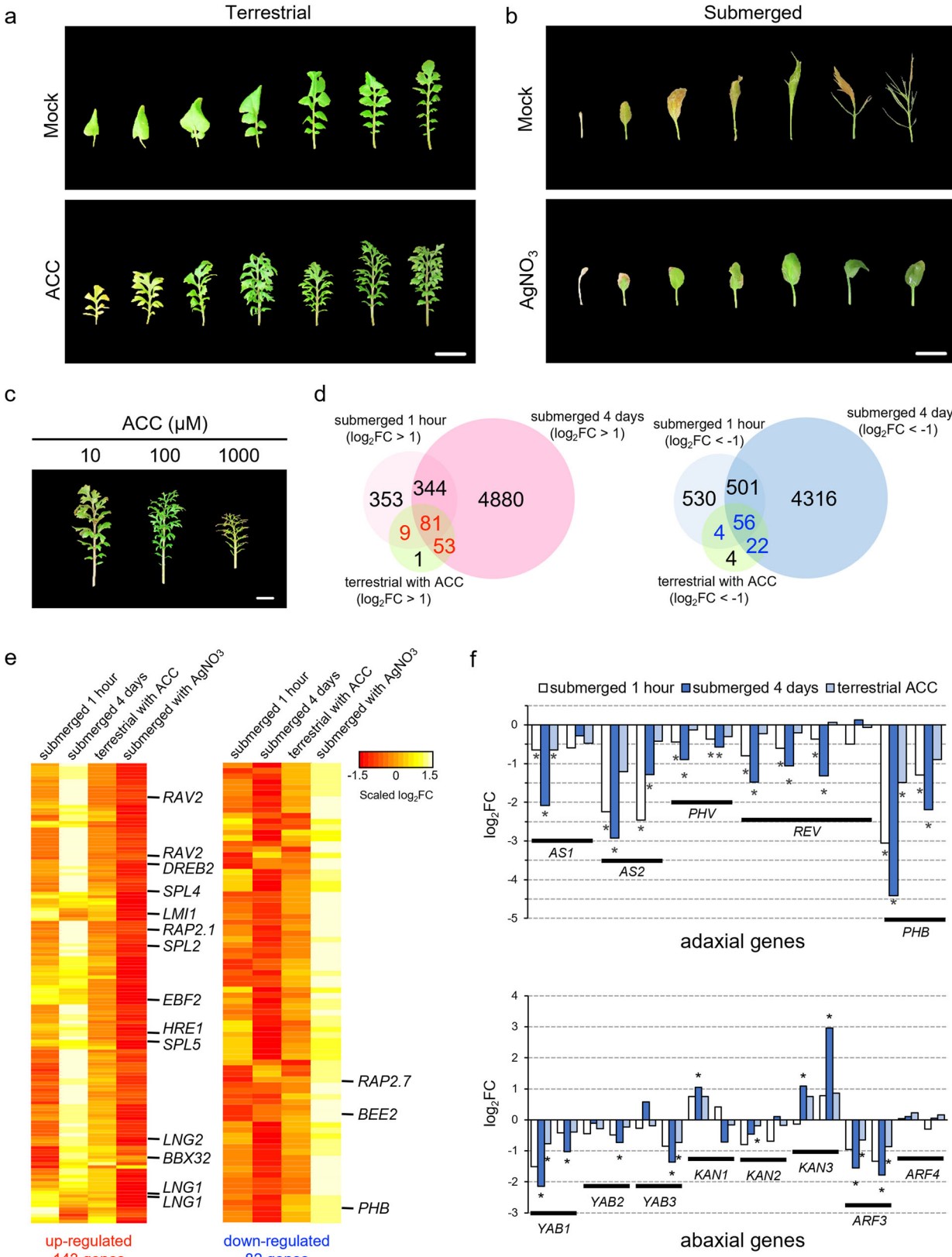

**Fig. 5 | Effect of ethylene on *Rorippa aquatica* heterophylly.** Leaves produced after treatment with **a** mock or 100 μM 1-aminocyclopropane-1-carboxylic acid (ACC, an ethylene precursor) under terrestrial conditions, or **b** mock or 1 μM AgNO₃ (ethylene inhibitor) under submerged conditions (scale bars, 1 cm). **c** Ethylene-dose effect under terrestrial conditions, with different ACC concentrations (scale bars, 1 cm). **d** Selection of candidate genes that induce submerged-phenotype leaves:

genes that were up- or down-regulated under either submergence or ethylene treatment (FDR < 0.01 and |Log₂FC| > 1) were used as candidates. **e** Candidate gene expression profiles. **f** Responses of adaxial–abaxial polarity determining genes to the submergence and ethylene treatments. "*" near plots indicates significant differences against control samples (FDR < 0.01).

**Fig. 6 | Effect of blue light on *Rorippa aquatica* heterophylly. a** Mature leaves grown under blue light conditions (left panel; scale bar, 1 cm); magnified view of leaflet (right panel; scale bar, 1 mm). **b** Leaf transcriptome profile under white and blue light conditions. **c** Mechanistic model for heterophylly in response to submergence.

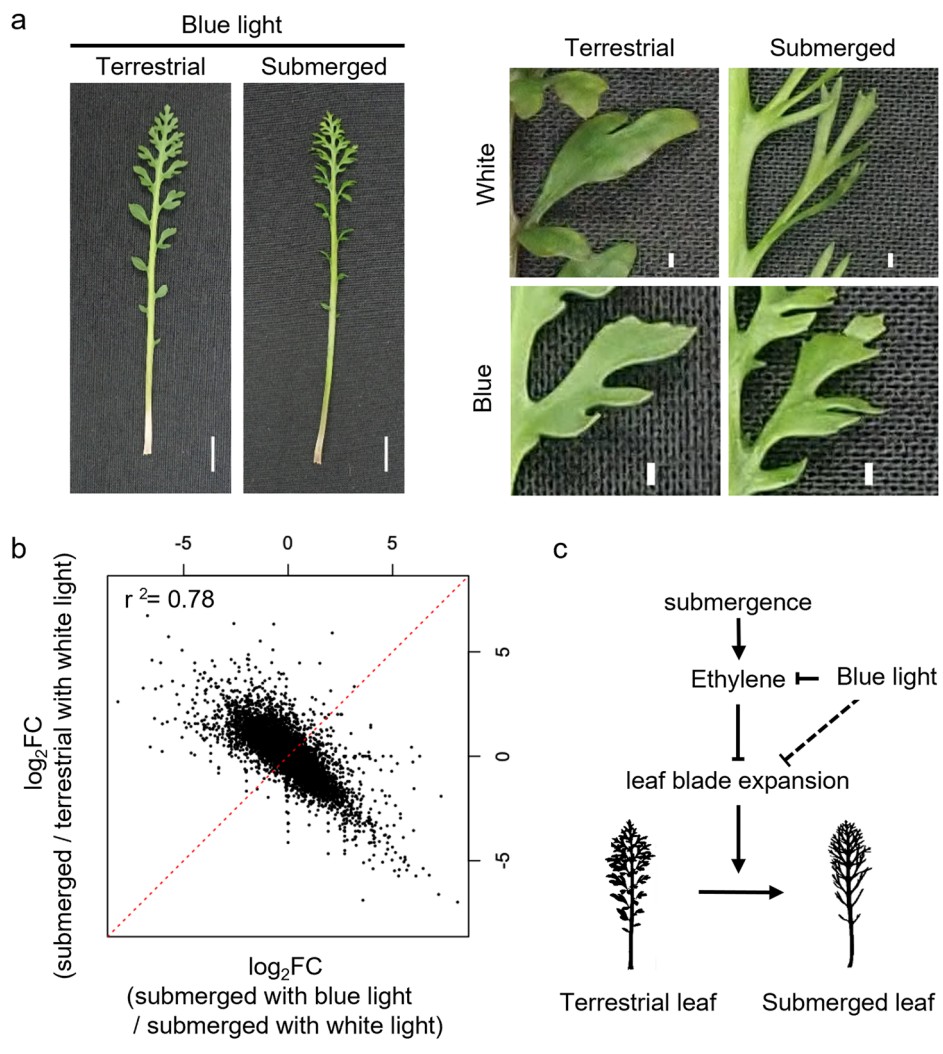

*PHABULOSA* (*PHB*), a member of the class III HD-ZIP gene family, directs cells to adaxialize. Its dominant mutant, *phb-1d*, forms adaxialized radial leaves[37]. Loss of function of *PHB* and other related class III HD-ZIP genes results in abaxialized radial cotyledons[38]. The establishment of adaxial–abaxial polarity is required for leaf blade expansion, and loss of this polarity leads to leaf radialization. The involvement of adaxial–abaxial polarity regulation in heterophylly was reported for *Ranunculus trichophyllus*[6]: submergence upregulated *KANADIs* genes which regulates abaxial growth, resulting in the formation of abaxialized radial leaves. In the present study, the expression of most adaxial genes, including *PHB*, were down-regulated under submergence and ACC-treated conditions (Fig. 5f). Regarding abaxial genes, one of the *KAN1* and *KAN3* paralogs were significantly upregulated after 4 days of submergence. These results suggested that loss of polarity to abaxialization may be involved in the formation of submerged-type leaves. Contrarily, other abaxial genes, *YABBYs* (*YABs*) and *ARF3* were downregulated in response to submergence.

**Blue light inhibits the submergence signal**
In addition to genes related to leaf morphogenesis, genes in the GO categories "response to light stimulus" and "shade avoidance" were also affected by submergence (Fig. 4b). Submergence alters light quality through absorption and reflection in water and light quality affects various physiological responses in plants. Therefore, we investigated the effect of light quality on leaf form under different light conditions. Blue light induced a pronounced response by elongating leaves along the anterior–posterior axis and preventing leaflet narrowing in response to submergence (Fig. 6a). The

Dissection Index (calculated from perimeter/square root of leaf area), as an indicator of leaf complexity, was subjected to statistical analysis (Supplementary Fig. 8). No significant difference in leaf complexity was detected between terrestrial leaves under white and blue light conditions. Under white light condition, submerged leaves showed high complexity with significant differences between terrestrial leaves. However, submergence under blue light caused an increase in complexity, and leaves with significant lower complexity than that under white light condition were formed. Our transcriptome analysis comparing the effects of white and blue lights revealed that the expression profile of the submerged leaves under blue light was negatively correlated with that under white light (Fig. 6b). This suggests that gene regulation normally induced during submergence under white light is not induced by submergence under blue light. Particularly, the expression of ethylene response genes induced by submergence decreased under blue light (Supplementary Fig. 9). In addition, the expression of genes regulating abaxial–adaxial polarity was not reduced under blue light. These results indicate that the submergence signal was inhibited under blue light conditions via the ethylene response pathway.

**Discussion**
Chromosome-level genome assembly and comparative genomic analysis reveal the genome structure of *R. aquatica* and shed light on its origin and evolution. We found that the genome of *R. aquatica* arose by allotetraploidization through hybridization between two ancestral *Rorippa* species (Fig. 3). Hybridization occurred no earlier than 4.2 Mya when *R. aquatica* subgenome A diverged from *R. islandica*. Thus,

the genome of *R. aquatica* arose by hybridization between two *Rorippa* genomes with eight pairs of chromosomes. WGD was followed by post-polyploid descending dysploidy (from $n = 16$ to $n = 15$) mediated by nested chromosome insertion (NCI), forming the fusion chromosome RaChr15. NCIs (reduction from $n = 16$ to $n = 15$) also occurred in some populations of the tetraploid *Cardamine pratensis*[39]. Both NCI events in *Rorippa* and *Cardamine* involved chromosome AK8/6 as the recipient chromosome, recombining with chromosome AK2 to form chromosome RaChr15 in *R. aquatica*, and chromosome AK5 in *C. pratensis*. An even more advanced post-polyploid descending dysploidy was documented in the tetraploid *C. cordifolia*, where chromosome number was reduced from $n = 16$ to $n = 12$ due to the formation of five fusion chromosomes[40]. Contrastingly, the closely related tetraploid genomes ($2n = 4x = 32$) of horseradish (*Armoracia rusticana*) and watercress (*Nasturtium officinale*) contain structurally conserved parental subgenomes, except for a 2.4-Mb long unequal translocation in watercress[41]. In Brassicaceae, polyploidization is not a rare event. Three allotetraploid *Brassica* species arose from the hybridization of two out of three diploid species[42–44]. In *Cardamine*, identification of polyploid origin were well proceeded with CCP and next-generation sequencing technology. These studies uncovered potential parental species of polyploids and their evolutionary process[39,45]. In *Rorippa* genus, several polyploidizations in species- and intraspecific population-level is suggested based on the variation of chromosome number[46,47]. Contrarily, their origin and evolutionary process are not identified yet. The genome sequence of *R. aquatica* provides a genome-wide reference for future studies on *R. aquatica*, the entire genus *Rorippa*, and tribe cardaminoid (Cardamineae) genomes.

Our transcriptome analysis of *R. aquatica*, based on whole-genome assembly data, provides three key insights into heterophylly (Fig. 6c). First, the submergence signal is transmitted via ethylene, and ethylene signaling inhibits leaf blade expansion. We found that ethylene signaling was induced by submergence and that exogenous ethylene resulted in narrower leaves even out of water. These responses were also observed in other amphibious plants[4,6,13]. Ethylene signaling is a conserved pathway among angiosperms[48], and ethylene is utilized as a submergence signal owing to its accumulation in response to submergence via low gaseous diffusion in water. This function of ethylene is consistent with its apparent role in regulating heterophylly in response to submergence in various plant species. ABA is reported as a correlated factor with ethylene response. The synthesis process and response pathway of ethylene and ABA regulate each other antagonistically[49]. In amphibious plants, up- and down-regulations of ethylene and ABA in response to submergence and inhibition of submergence responses by exogenous ABA treatment were reported[4–6,13]. In *R. aquatica*, ABA responsive genes were highly enriched in downregulated DEG at early response to submergence and exogenous ABA treatment inhibited submergence-responded heterophylly. Due to complex relationships between ethylene and ABA, the function of ABA in submergence-related heterophylly have not been elucidated yet. Further analyses are essential to demonstrate whether ABA works as an inhibitor of ethylene or plays an independent role in submergence responses.

Gibberellins are also involved in regulating leaf form in *R. aquatica*, both under terrestrial and submerged conditions[3]. Gibberellin treatment resulted in the emergence of simple leaves at low temperatures and submergence, conditions that normally result in dissected leaves, and inhibition of gibberellin signaling resulted in dissected leaves even at high temperatures that normally induce simple leaves. While the effect of gibberellin content for morphology were identified, its molecular responses to submergence were unknown. This study showed the change of genes related to the gibberellin stimulus in response to submergence. GO analysis indicated the enrichment of gibberellin responsive genes in upregulated DEG as throughout and late responses and downregulated DEG as early response. The expression profile seems to increase gibberellin signaling in response to submergence at least in the late response. Gibberellins showed different effects in other amphibious plants that do not show apparent heterophylly in response to temperature under terrestrial conditions. In *Hygrophila difformis* and *Callitriche palustris*, inhibition of gibberellin signaling suppresses the formation of the submerged-leaf phenotype, and gibberellin treatment fails to induce this leaf phenotype under terrestrial conditions[4,13]. This suggests that leaf shape in *R. aquatica* may be regulated by two parallel pathways: submergence-responsive heterophylly mediated by ethylene and temperature-dependent heterophylly mediated by gibberellins. How the two pathways are regulated and interact is a topic for future work.

Our second important finding is that in *R. aquatica*, most adaxial polarity related genes were downregulated and some *KANs* of abaxial genes were upregulated under submergence. These results suggested that leaves narrowed down through abaxialization in response to submergence similar to that in *Ranunculus trichophyllus*[6]. However, no obvious morphological abaxialization, such as radialization of vein in *R. aquatica* submerged leaf, was observed in a previous study[3], which is similar to that in *Hygrophila difformis*. Its submerged leaves having narrow lamina kept normal polarity in vein and development of palisade tissue in adaxial side[4]. The establishment of adaxial–abaxial polarity eventually leads to the establishment of the middle domain, which is located at the juxtaposition between the adaxial–abaxial domains and participates in the leaf lamina outgrowth[50]. The radialized narrow leaves shown in *A. thaliana* polarity gene mutants[37] and submerged *Ranunculus trichophyllus*[6] developed only main vein. Contrarily, highly dissected submerged leaves in *R. aquatica* developed lateral veins. Temporal and spatial regulation of leaf polarity may form dissected complex leaves in *R. aquatica*.

A third important finding of our study is that blue light is involved in regulating heterophylly. At the gene expression level, blue light blocked the upregulation of ethylene response genes during submergence. A similar effect of blue light was observed in submergence responses of stomatal development in *R. aquatica*[33]. Submergence with white or red light promotes the expression of ethylene synthesis-related genes and inhibits stomatal development. Contrarily, these submergence responses were inhibited under blue light. Although the mechanism by which blue light regulates heterophylly through ethylene is unclear, the relationship between blue light and ethylene in the shade avoidance response has been studied. For example, in *A. thaliana*, low blue light induces stem elongation for shade avoidance, although not in ethylene-insensitive mutants[51]. The molecular mechanisms underlying the response pathways to blue light and ethylene are unclear, even in *A. thaliana*. In plants, blue-light reception is mediated by the cryptochromes (CRYs)[52]. In *Brassica napus*, overexpression of *CRY1* leads to downregulation of *1-aminocyclopropane-1-carboxylate synthase 5* and *8* genes associated with ethylene biosynthesis[53]. Changes in light quality and quantity do not seem to be the main factors of submergence signals, owing to only a small difference being caused by experimental submergence treatment[33]. However, under natural conditions, the quality of light reaching submerged plants changes with water level and condition due to differences in the light-absorbance ratio. Therefore, changes in light quality could provide coordinator signals about submerged conditions. Our results show that blue light plays an important role in regulating heterophylly in response to submergence. Considering that the amphibious fern *Marsilea quadrifolia* showed a similar response[54], blue light signaling may also be central to heterophylly in various plants.

Paralogs related to ethylene response or adaxial–abaxial polarization showed similar expression patterns in response to environmental stimuli (Fig. 5f and Supplementary Fig. 8). This suggests that a change in function or expression in the upstream regulatory gene causes the heterophylly upon submergence. The allotetraploid origin of *R. aquatica* suggests several possible mechanisms by which the species acquired traits such as amphibiousness and heterophylly in response to various signals. A most simple hypothesis is the inheritance from either parent. However, this possibility is quite low because only *R. aquatica* shows amphibiousness and remarkable heterophylly in *Rorippa* genus. As another possibility, the hybridization of two *Rorippa* species at the first step of allotetraploidization may cause enhanced function like heterosis. The most likely hypothesis is the achievement of new functions caused by WGD. Furthermore, the redundancy of duplicated genes often enables genes to acquire novel functions. Finally, accelerated accumulation of mutations could have led to specific

traits. Future comparative studies in the genus *Rorippa* will determine the origin of *R. aquatica* subgenomes and elucidate the evolution of adaptive mechanisms in a submerged condition.

## Methods

### Plant materials

*Rorippa aquatica* plants (two accessions, N and S)[18] were kept in a growth chamber at 30 °C under continuous light at 50 μmol photons m$^{-2}$ s$^{-1}$ supplied by a fluorescent lamp. For each treatment, plants that regenerated from a leaf tip as described earlier[19] were used.

To induce inflorescences for comparative chromosome painting, the growth chamber temperature for *R. aquatica* accession S was changed to 20 °C. Young inflorescences were collected from plants and fixed in freshly prepared fixative (ethanol: acetic acid, 3:1) overnight, transferred to 70% ethanol, and subsequently stored at −20 °C.

For sequencings of genome, *R. aquatica* accession N plants were grown in a growth chamber at 30 °C and used to extract genomic DNA.

For morphological and transcriptome analyses, *R. aquatica* accession N plants were used. All plants used for these analyses were grown in a growth chamber at 25 °C. Plants used to examine the change of morphology and gene expression after the transition to the submerged condition were grown in glass tanks with an approximate 8 cm water depth. To examine effects of ethylene on heterophylly, the plants were treated with 100 μM 1 aminocyclopropane-1-carboxylic acid (ACC) or 1 μM AgNO$_3$ by supplement of water containing chemical. For chemical treatment under submerged conditions, the chemicals to be tested were diluted in the 200 mL sterile distilled water in the culture jar. For morphological analysis, plants were grown under each treatment for two months until the leaves were mature. To examine the effect of ethylene amount, plants were treated with different concentrations of ACC (10, 100, and 1000 μM) under the terrestrial condition described above. To examine effects of ABA on heterophylly, the plants were transfer to submerged condition containing 10 μM ABA and grown for 3 weeks. To examine the effects of blue light, plants were grown under 20 μmol photons m$^{-2}$ s$^{-1}$ supplied by a blue LED. For RNA-seq analyses, plants grown under terrestrial condition with white light at 25 °C were transferred to each experimental condition and grown at certain period (ACC: 1 day, AgNO$_3$: 1 h, blue or white light: 1 h), and subjected to RNA extraction.

### Chromosome preparation

Chromosome spreads from fixed young flower buds containing immature anthers were prepared according to published protocols[55,56]. Chromosome preparations were treated with 100 μg/mL RNase in 2× sodium saline citrate (SSC, 20× SSC: 3 M sodium chloride, 300 mM trisodium citrate, pH 7.0) for 60 min, and with 0.1 mg/mL pepsin in 0.01 M HCl at 37 °C for 5 min, then post-fixed in 4% formaldehyde in distilled water and dehydrated via an ethanol series (70%, 90%, and 100%, 2 min each).

### Painting probes

For comparative chromosome painting (CCP), 674 chromosome-specific BAC clones of *Arabidopsis thaliana* (The Arabidopsis Information Resource, TAIR; http://www.arabidopsis.org) were used to establish contigs corresponding to the 22 genomic blocks (GBs) and eight chromosomes of the Ancestral Crucifer Karyotype (ACK)[24]. To determine and characterize inversions of GBs on chromosome Ra15, BAC contigs corresponding to GBs D and E were split into smaller subcontigs and differentially labeled and used in several consecutive experiments. All DNA probes were labeled with biotin-dUTP, digoxigenin-dUTP, or Cy3-dUTP by nick translation as described earlier[57].

### Comparative chromosome painting

DNA probes were pooled appropriately, ethanol precipitated, dried, and dissolved in 20 μL of 50% formamide and 10% dextran sulfate in 2× SSC. The dissolved probe (20 μL) was pipetted onto a chromosome-containing slide and immediately denatured on a hot plate at 80 °C for 2 min.

Hybridization was conducted in a moist chamber at 37 °C overnight. Post-hybridization washing was performed in 20% formamide in 2× SSC at 42 °C. Hybridized probes were visualized either as the direct fluorescence of Cy3-dUTP or via fluorescently labeled antibodies against biotin-dUTP and digoxigenin-dUTP[57]. Chromosomes were counterstained with 4′,6-diamidino-2-phenylindole (DAPI, 2 μg/mL) in Vectashield antifade (Vector Laboratories). Fluorescence signals were analyzed and photographed using a Zeiss Axio Imager epifluorescence microscope with a CoolCube camera (MetaSystems, Altlussheim, Germany). Images were acquired separately for all four fluorochromes using appropriate excitation and emission filters (AHF Analysentechnik, Tübingen, Germany). The four monochromatic images were pseudocolored, merged, and cropped using Photoshop CS (Adobe Systems, Mountain View, CA) and ImageJ (National Institutes of Health, Bethesda, MA).

### Illumina genome DNA sequencing

Genome-seq libraries were constructed using whole- or nucleic-genome DNA. For extraction of nucleic DNA, the nuclear fraction was prepared from whole plants using the 'Semi-pure Preparation of Nuclei Procedures' protocol of the CelLytic PN Isolation/Extraction Kit (Sigma-Aldrich, St. Louis, MO). Next, genomic DNA was isolated from the nucleus or whole plant using a DNeasy Plant mini kit (Qiagen, Hilden, Germany). Genome-seq libraries were prepared using the Nextera DNA Sample Prep Kit. Sequencing was performed using NextSeq 500, generating paired-end reads of 151 bp.

### PacBio genome DNA sequencing

DNA for PacBio library was prepared as follows: crude nuclei were obtained from regenerated plants (using ca. 1 cm lengths of leaf tip) using the 'Crude Preparation of Nuclei Procedures' protocol of the CelLytic PN Isolation/Extraction Kit (Sigma-Aldrich). DNA extraction from crude nuclei was performed using two different methods. For the first run, the Dneasy Plant mini kit was used. For the subsequent two runs, genomic DNA was extracted using phenol/chloroform/isoamyl alcohol extraction with CTAB buffer and purified using QIAGEN Genomic-tip 20/G. Long reads were generated using the PacBio RS II system.

### Hi-C Seq

Hi-C Seq sample was prepared following a method reported in an earlier study[58]. HindIII was used for DNA digestion. For the preparation of the sequencing library, the purified Hi-C sample (500 ng) was diluted to 500 μl with dH$_2$O, and 500 μl of 2× binding buffer (BB) (10 mM Tris, 1 mM EDTA, 2 M NaCl) was added. The diluted Hi-C samples were fragmented to a mean size of 300 bp by sonication using a Covaris M220 sonication system (Covaris, Woburn, MA, USA) in a milliTUBE 1 ml AFA Fibre (Covaris). The parameters of the program were as follows: power mode, frequency sweeping; time, 20 min; duty cycle, 5%; intensity, 4; cycles per burst, 200; temperature (water bath), 6 °C. Biotin-labeled Hi-C samples were then enriched using MyOne Streptavidin C1 magnetic beads (Veritas, Tokyo, Japan). For this, 60 μl of streptavidin beads were washed twice with 400 μl of Tween Wash Buffer (TWB) (5 mM Tris, 0.5 mM EDTA, 1 M NaCl, 0.05% Tween-20). The recovery of streptavidin beads was performed by placing the tubes on a magnetic stand. Subsequently, the beads were added to 1 ml of sheared Hi-C sample. After 15 min of incubation at room temperature under rotation, the supernatant was removed, and the beads binding biotinylated Hi-C fragments were resuspended in 400 μl of 1× BB. Then, the beads were washed once in 60 μl RSB (Resuspension buffer) (Illumina, San Diego, CA, USA) and resuspended in 50 μl RSB. The enriched biotinylated DNA fragments were subjected to library construction on beads using the KAPA HyperPrep Kit for Illumina (Roche, Basel, Switzerland) according to the manufacturer's protocol, with 18 cycles of PCR for library amplification. The amplified DNA fraction (50 μl) was corrected and purified using Agencourt AMPure XP (Beckman Coulter) following the standard protocol and finally resuspended in 15 μl of RSB. The library was sequenced using a NextSeq 500 system, generating paired-end reads of 151 bp.

## Genome size estimation

The genome size of *R. aquatica* was estimated by k-mer counting using jellyfish2 (http://www.genome.umd.edu/jellyfish.html). K-mers from Illumina read data were counted, and the k-mer distribution was plotted; the distribution peaks from homozygous regions were picked manually, and genome size (in bases) was calculated as a total number of k-mers/peak of the k-mer distribution.

## Genome assembly and annotation

Genome assembly was performed using MaSuRCA[26] with both Illumina and PacBio reads. The assembled scaffolds were error-corrected using Pilon[59]. Scaffolding into chromosome-level sequences was performed via the 3D de novo assembly (3D-DNA) pipeline[60], using the assembled scaffolds and Hi-C Seq reads. The remaining gaps in chromosome-level sequences were filled by LR_Gapcloser[61], using PacBio reads that were error-corrected using ColorMap[62]. Assembled genome sequences were benchmarked using Benchmarking Universal Single-Copy Orthologs (BUSCO)[63] with a land-plant dataset (embryophyta_odb9). Repeat sequences in the genome were identified and masked using RepeatModeler and RepeatMasker (http://www.repeatmasker.org). Gene prediction was performed using the PASA pipeline[28]. Three types of prediction were used: 1) ab initio prediction using AUGUSTUS[64], GlimmerHMM[65], and SNAP, with an *Arabidopsis* training dataset; 2) Protein homology detection using EXONERATE with *A. thaliana* TAIR10 protein data; and 3) Alignment of assembled transcripts to the genome. Transcriptome data were obtained by de novo assembly using Trinity[66], with RNA-seq data (DRA006777) from an earlier study[19]. All the predicted gene structures were integrated into the final gene data using EvidenceModeler (EVM)[67] and PASA. Gene Ontology terms were assigned to each transcript using Blast2GO[68] based on the results of a BLASTP homology search against the non-redundant protein sequence (Nr) database and InterProScan.

## Genome structure

Alignment of *R. aquatica* chromosome sequences to the *C. hirsuta* genome was performed using MUMMER[69]. Genome structure (distribution of genes, long terminal repeats, and links between paralogous genes) was illustrated using CIRCOS[70].

## Comparative genomics analysis

We performed whole-genome level phylogenetic analysis using *R. aquatica* genomic information and genome-level data of several plant species. The protein dataset of 22 plant species were obtained from the Phytozome database (https://phytozome-next.jgi.doe.gov/). The datasets of *C. hirsuta* and *Barbarea vulgaris* were prepared using data from the genome database of each species (http://chi.mpipz.mpg.de/[71] and https://www.ebi.ac.uk/ena/browser/view/LXTM01000000[72], respectively). We prepared draft *Rorippa islandica* genomic data via genome assembly using Velvet[73] with genome-seq read data (SRR1801303) from the Sequence Read Archive, setting k-mer to 151. Using an Arabidopsis training dataset, gene prediction was performed using AUGUSTUS[74]. Protein sequences of a single representative longest transcript variant for each gene were extracted using an inhouse Perl script. Using OrthoFinder[75], each protein sequence was clustered into an orthogroup based on similarity, and the phylogenetic analysis of each orthogroup was integrated as a species tree.

The synonymous substitution rate (Ks) was calculated to estimate evolutionary event ages. Using MACSE[76], we performed multiple-alignment of the coding DNA sequences in each orthogroups in which single-copy conserved genes in seven related Brassicaceae species (*Eutrema salsugineum*, *Arabidopsis thaliana*, *Arabidopsis lyrata*, *Barbarea vulgaris*, *Cardamine hirsuta*, *Capsella rubella*, and *Rorippa islandica*) and duplicated genes in *R. aquatica* were classified, and then we calculated Ks using yn00 in the PAML package (http://abacus.gene.ucl.ac.uk/software/paml.html). The age of each event was estimated as T (in years) = Ks of peak/(2 * $\mu$), where $\mu$, which is the synonymous divergence rate per site per year, equals 6.51648E−09 in Brassicaceae[32].

## Transcriptome analysis

Total RNA was isolated from shoot apexes containing young leaves using a RNeasy Plant Mini kit (QIAGEN, Hilden, Germany). RNA-seq libraries were prepared using the Illumina TruSeq Stranded RNA LT kit (Illumina, CA, USA), according to the manufacturer's instructions. Libraries were sequenced on the NextSeq500 sequencing platform (Illumina, CA, USA), and 76 bp single-end reads were obtained. The reads were mapped to the genome sequences of *R. aquatica* using Tophat2. Count data were subjected to a trimmed mean of M-value normalization in the edgeR[77]. Transcript expression and DEGs were defined using the edgeR GLM approach with cut off threshold, false discovery rate (FDR) < 0.01, and |log2FC| > 1. Summary of RNA-seq read data and mapping are shown in Supplementary Data 5.

## Statistical analysis of leaf complexity

To assess complexity of leaf form, Dissection Index was calculated from morphological image using the following formula (perimeter / square root of leaf area). Perimeter and area of leaves were measured from photographs using ImageJ software[78]. Statistical analysis was carried out using Tukey-Kramer test.

## Statistics and reproducibility

The information of statistical tests used in the study are provided in the respective methods sections and Supplementary Data. The RNA-seq was performed in three independent biological replicates and statistically analyzed using the edgeR GLM approach. For leaf complexity, 8–21 leaves from each experimental condition were measured and mean dissection indexes were statistically analyzed using Tukey-Kramer test.

## Reporting summary

Further information on research design is available in the Nature Portfolio Reporting Summary linked to this article.

## Data availability

The assembled *R. aquatica* genome sequences and their annotations were deposited in Figshare (10.6084/m9.figshare.19207362). The draft assembly data of *R. islandica* were deposited in Figshare (10.6084/m9.figshare.24849201). Genome-seq read data and Hi-C seq read data are available in the DDBJ Sequenced Read Archive (DRA) under the accession numbers DRA010675 and DRA013596, respectively. Transcriptome read data are also available in DRA under DRA014113, DRA014114, DRA014164, and DRA014165. The numerical source data behind the graphs in Figs. 3a and 5f can be found in Supplementary Data 6. All other data are available from the corresponding author on reasonable request.

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

## Acknowledgements
We thank Dr. Neelima Sinha, Dr. Naomi Nakayama, and Dr. Dhanya Radhakrishnan for valuable discussions. This work was financially supported by the JSPS KAKENHI grants 21H02513 and MEXT-Supported Program for the Strategic Research Foundation at Private Universities grant S1511023 to S.K. and JSPS KAKENHI grants 20H05911 and 22H00415 to S.M. This work was also supported by the National Key Research and Development Program of China (2017YFE0128800), the National Natural Science Foundation of China (31870384, 32101254), and the International Partnership Program of the Chinese Academy of Sciences (152342KYSB20200021) to H.H. and G.L., and by the project TowArds Next GENeration Crops (no. 17 CZ.02.01.01/00/22_008/0004581) of the ERDF Programme Johannes Amos Comenius to M.A.L and T.M.

## Author contributions
To.S., S.I., H.N., T.M., and Ta.S. performed the experiments; To.S. and S.K. carried out the bioinformatic analyses; To.S., H.N., T.M., G.G., Ta.S., G.L., M.A.L., and K.S. wrote the article with input from all authors; H.H., S.M., M.A.L., and K.S. supervised; all authors discussed and commented on the article.

## Competing interests
The authors declare no competing interests.
