## [Peer review file · Communications Biology]

nature portfolio

Peer Review File

~~**Open Access** This file is licensed under a Creative Commons Attribution 4.0 International License, which permits use, sharing, adaptation, distribution and reproduction in any medium or format, as long as you give appropriate credit to~~

~~the original~~ author(s) and the source, provide a link to the Creative Commons license, and indicate if changes were made. In the cases where the authors are anonymous, such as is the case for the reports of anonymous peer reviewers, author attribution should be to 'Anonymous Referee' followed by a clear attribution to the source work. The images or other third party material in this file are included in the article's Creative Commons license, unless indicated otherwise in a credit line to the material. If material is not included in the article's Creative Commons license and your intended use is not permitted by statutory regulation or exceeds the permitted use, you will need to obtain permission directly from the copyright holder. To view a copy of this license, visit <http://creativecommons.org/licenses/by/4.0/>.

Reviewers' comments:

Reviewer #1 (Remarks to the Author):

This work reported the genome of one Cruciferae species with heterophylly. They recovered this species experienced a recent tetraploidization through hybridization. Especially, after this whole-genome duplication, two chromosomes fused to produce the decreased chromosome number. Using genomic data and transcriptome analyses, authors further found that ethylene signaling contributed greatly to heterophylly.

The total ms was well written although there is no exciting discovery. My major concerns comprise three points:

1. The recent allotetraploid origin of *Rorippa aquatica* did not give strong evidence and related analyses. I suggest to add more transcriptomes or genome-resequencing data to construct gene trees. Using gene tree topologies support the hybrid allopolyploid origin.
2. One chromosomal fusion between two chromosomes is the most interesting. It is necessary to explore whether this fusion is related to heterophylly. Therefore, authors need to check whether related genes on the fusion point are involved in ethylene.
3. It is better to write one summary about allotetraploid hybridization in the family Cruciferae. In fact, such speciation is very common in the family.

Reviewer #2 (Remarks to the Author):

In this manuscript, the authors demonstrate that *R. aquatica* evolved through allopolyploidization, as revealed by comparative chromosome painting, and through modification of ethylene signaling, as revealed by RNA-seq analysis.

Given the scarcity of molecular studies on amphibious plants, this work is a valuable resource for future research. Understanding how plants adapt to aquatic environments is particularly important in the face of urgent global issues such as eustatic sea level rise and the need for sustainable agriculture.

As a model system for heterophilic plants, the authors analyzed *R. aquatica*, a close relative of *Brassica*. This approach is clever because it allows the use of knowledge accumulated from *Arabidopsis* and *Cardamine* species for chromosome analysis as well as transcriptome analysis. Especially, the authors proved the beauty of such approach for chromosome analysis and clearly show that *R. aquatica* is evolved through allopolyploidization.

However, some aspects of the RNA-seq analysis need to be confirmed. The authors suggest that GA signaling affects on the heterophylly from RNA seq analysis. But this conclusion is just based on Nakayama et al's paper (Plant Cell, 2014). The RNA seq data in Fig 4B show that 'GA response' is down-regulated at early response but is up-regulated at late response for submergence. We cannot determine whether GA induces terrestrial leaves or aquatic leaves without additional information from Nakayama et al's paper. Similarly, 'ABA response' is down-regulated early on, but up-regulated later. Then, how can we interpret this data? Can't we interpret that ABA also induces terrestrial leaves? Moreover, the 'ABA response' is much stronger than 'GA response' during early response to submergence. In terms of the evolutionary viewpoint, it is also consistent with the role of ABA, a plant hormone evolved to adapt to terrestrial environment. Therefore, I feel it should be confirmed

that if ABA also induces terrestrial leaves. The experiment required for this confirmation is simple and quick, so adding this data would be beneficial.

I disagree with the authors' interpretation of the expression of polarity genes. Although they suggest that all polarity genes are reduced, their results in Fig 5F contradict this interpretation. While the expression of all the adaxial genes is indeed repressed, the expression of abaxial genes is not simply repressed. Instead, the level of KANADI3 (KAN3) is increased after submergence or ethylene treatment. This result is consistent with Kim et al's paper (PloS Genet, 2018) showing that KAN3, the major KAN gene expressed higher than KAN1 or KAN2. It is possible that KAN3 is also the major adaxial gene in *R. aquatica*. In that case, because of genetic redundancy and dominance of KAN3, changes in leaf polarity may also affect heterophylly of *R. aquatica*. This can be easily checked in the RNA seq data by directly comparing the absolute value of KANs.

Reviewer #3 (Remarks to the Author):

This manuscript is on the amphibious plant *Rorippa aquatica* with an interesting adaptational mechanism to submergence. It has two topics. The first part contains the assembly and annotation of the *Rorippa aquatica* genome, which has a rather complex evolutionary history. The second part is a transcriptomic study of the molecular basis for heterophylly in this species under submergence. Both parts have an importance for the scientific community.

While the first part is well-documented and new with only some smaller questions remaining, the second part has some major problems. One part of the problems comes from missing information on the experiments and poor presentation and availability of data. The other part comes from the fact that a portion of the data has been already used in a previous publication, and this is not clearly stated in the manuscript. Below I will specify my comments.

Comments on part 1 (genomics)

As far as I could learn from literature, the species *Rorippa aquatica* is usually propagated clonally, even in nature. It would be good if you could add this information into the introduction since it is relevant. Did you get any hints from your genomic data and experiments whether the plant is able to produce seeds, and if not, what is the reason for it? It would be also good to have information on whether the chromosome pairs (in the diploid stage) are identical, i.e. homozygous as is the case for *Arabidopsis*, or whether they are heterozygous. If the species produces seeds, does this require different individuals, i.e. is it a non-selfing species?

In the introduction, I would like to suggest more references. For example, the second paragraph, lines 31 to 43, does not contain any reference, also the statement in lines 65-67.

In discussion, it would be also good to include some facts on the conditions under water, i.e. low gas diffusion. This is the major factor that causes ethylene accumulation and therefore is the first signal for submergence in most plant species. This should be made clearer in the discussion, lines 352 to 357.

Why did you not add *Nasturtium officinale* as a species into the comparisons? It should be even more related to *Rorippa* than *Barbarea* species. The genome sequence is not well annotated, but has

been published here: Kiefer et al. 2019, Nature Plants.

Supplementary Figure 5, could you use two other colors for the *Rorippa* species? These two colors can be hardly distinguished from the *Eutrema* and *Arabidopsis* data.

The link to the *Barbarea* genome data is no longer functional (line 559), please check and provide a correct link.

Please provide data on the species comparison, for example with a supplemental table listing all orthologs of all species, at least for the 10,845 genes that you used for further analyses (line 167). Only in this way other researchers can use your dataset.

Comments on part 2 (transcriptomics)

As far as I can see from the raw data IDs, part of the data have been used in another paper, presumably by the same group, namely the 1h and 4d time point, and the different light treatments after 1h. It is fine to use the data in another publication, but it should be clearly stated.

Furthermore, it remains unclear which data have been added newly here, since the DRA IDs are not yet accessible. Please provide a Supplementary Table with read counts, mapping statistics and conditions for the RNAseq data used here, as you did in the other paper.

Moreover, please also provide the full dataset. You identified more than 9,000 DEGs, but here you provide only the expression data for 143 up- and 82 down-regulated genes. At least, you should give the expression data (log2FC, P-value, cpm) for the DEGs with ACC and AgNO₃ treatment.

It is unclear how the treatment was actually done, and how long after treatment you harvested the tissue for RNAseq. The method part does not mention details in line 578. In line 435 you describe a treatment for two months, but this probably was used for making the pictures in Fig. 5. In addition, how did you apply the ACC, and how often? Did you spray it onto the plants? Did you spray only once? What was the composition of the spraying solution?

In line 232 and 585, you do not mention the cut-off that you used to define DEGs. Probably it is the same as in the previous paper.

As far as I understood, you used also AgNO₃ treatment under submergence, but you do not present the data in Figure 5 or in the text. Why did you not comment on those data? Did they not fit into your hypothesis?

Section on LMI1 and RCO (line 290 to 298), there is another RaRCO copy and two RaLMI1 copies. Do they behave similarly or differently?

Figure S6 and section on blue light, is this the same dataset as in the previous paper, or a different one, for example a different time point? Please provide more details and give numbers of DEGs/ GOs and so on. A high percentage of genes that were shown in this figure do not meet the selection criteria of log2FC >1 or <-1. It is therefore doubtful whether the impact of blue light has such a big impact on this set of genes. So far, statistics are missing (see also next comment).

In Figure 5F and S6, you need to add statistics and standard deviations for the log2FC values. I

assume that those data are from the RNAseq dataset and not from subsequent qRT PCR analyses?
And the data from the AgNO₃ treatment are missing again.

Fig. 4A and 6A, can you provide quantitative data for your observations? With one picture from one leaf, one cannot conclude a lot. It should be easy to measure leaf thickness or area for Figure 6.

Fig. 5B, the mock submergence treatment looks quite strange, compared with Fig. 1B. Can you add details to the legend about length of treatment?

Line 363 to 371 on GA, which genes are you referring to? Can you give a list or mention IDs? And you should mention the species names for which GA had no impact.

Line 395 to 399, please provide more details on light quality changes under water in nature. However, in your water depth, the changes in light quality might be minor.

There are some minor errors in the text, for example line 266, is the word "various" correctly used in this context; line 269, "one" should be "1"; line 316, remove "-".

Dear reviewers

We highly appreciate you reviewing our manuscript (COMMSBIO-23-0769-T) entitled "Chromosome-level genome assembly of the model amphibious plant *Rorippa aquatica* reveals its allotetraploid origin and mechanisms of heterophylly upon submergence".

We took all the comments seriously and performed additional experiments during the revision. In accordance with the reviewer's comments, we revised the main text and figures. Highlighted texts in the revised manuscript indicate modified points. The major changes are summarized below.

- We added statistical analysis data to Fig. 5F and Supplementary Fig. 8 (Supplementary Fig. 6 in the previous version).
- Supplementary Fig. 5 was replaced with a modified version to improve visibility.
- A new supplementary figure was added as Supplementary Fig. 6, to show chromosome level Ks distributions between *B. vulgaris* and *C. hirsuta* in addition to *R. islandica*.
- A new supplementary figure showing statistical analysis of leaf complexity was added as Supplementary Fig. 7.
- The following 3 Supplementary Dataset is added.
 - Supplementary Dataset 1. Ortholog clustering in *Rorippa* and Brassicaceae.
 - Supplementary Dataset 2. RNA-seq Expression profile data.
 - Supplementary Dataset 5. Information of RNA-seq read and mapping statistics.

Please find below our detailed responses to the reviewers' comments. We hope that our revised version addresses all of the concerns raised. We would, of course, be willing to make additional revisions to make the manuscript acceptable for publication.

Responses to reviewer #1

Comment 1. The recent allotetraploid origin of *Rorippa aquatica* did not give strong evidence and related analyses. I suggest to add more transcriptomes or genome-resequencing data to construct gene trees. Using gene tree topologies support the hybrid allopolyploid origin.

Response 1. Thank you for your valuable comment. Ks showed phylogenetic distance among orthologous gene pairs and the distribution of Ks between the whole ortholog pair from *R. islandica* genome and each chromosome of *R. aquatica* show chromosome-level phylogenetic distance. It is more reliable than a phylogenetic tree from a single or a few numbers of genes. However, we added chromosome level Ks analyses against *Barbarea vulgaris* and *Cardamine hirsuta* (new Supplementary Fig. 6) to support our hypothesis. Ks against these species don't show clear segregation, indicating all *R. aquatica* chromosomes are located in approximately the same phylogenetic distance against these 2 species.

Comment 2. One chromosomal fusion between two chromosomes is the most interesting. It is necessary to explore whether this fusion is related to heterophylly. Therefore, authors need to check whether related genes on the fusion point are involved in ethylene.

Response 2. We are also concerned what change in the genome caused heterophylly. However, as mentioned in the Discussion, many possible causes exist, such as inheritance from mutated parent species, heterosis, structural alternation of the genome including chromosome fusion, and mutation after species deviation. WGD and the following event affected too many genes. As BUSCO analysis showed only 57.3% duplication in the conserved gene, it might indicate loss of either one of the paralogous genes after WGD independent of chromosome structural alternation. In addition, it is difficult to assess the effect of the elimination of a single gene due to the lack of gene manipulation methods in this species. Because a duplicated genome has redundancy, only showing a lack or mutation in one gene is insufficient to demonstrate its involvement in specific phenomena.

We will try to identify whether heterophylly is caused by chromosome fusion or not as future work.

Comment 3. It is better to write one summary about allotetraploid hybridization in the family Cruciferae. In fact, such speciation is very common in the family.

Response 3. Thank you for your valuable suggestion. We added a summary about polyploidization in Brassicaceae in Discussion (Lines 365-372).

Response to reviewer #2

Comment 1. However, some aspects of the RNA-seq analysis need to be confirmed. The authors suggest that GA signaling affects on the heterophylly from RNA seq analysis. But this conclusion is just based on Nakayama et al's paper (Plant Cell, 2014). The RNA seq data in Fig 4B show that 'GA response' is down-regulated at early response but is up-regulated at late response for submergence. We cannot determine whether GA induces terrestrial leaves or aquatic leaves without additional information from Nakayama et al's paper. Similarly, 'ABA response' is down-regulated early on, but up-regulated later. Then, how can we interpret this data? Can't we interpret that ABA also induces terrestrial leaves? Moreover, the 'ABA response' is much stronger than 'GA response' during early response to submergence. In terms of the evolutionary viewpoint, it is also consistent with the role of ABA, a plant hormone evolved to adapt to terrestrial environment. Therefore, I feel it should be confirmed that if ABA also induces terrestrial leaves. The experiment required for this confirmation is simple and quick, so adding this data would be beneficial.

Response 1. Thank you for your suggestion. As you pointed out, genes that respond to ethylene, gibberellin, ABA, and auxin were enriched in DEG with submerged treatment. Pathways related to ABA (and auxin) may be involved in submergence responded heterophylly. In this study, we focus on ethylene and gibberellin by following reasons in addition to the results of GO enrichment analysis. Because ethylene is thought as most important submergence signal in various amphibious plants and gibberellin works as a main regulator of temperature-dependent heterophylly in *R. aquatica* based on past studies, we must mention them. And, one of the purposes of this paper is the construction of the genome information basis of *R. aquatica* and the assessment of transcriptome analysis with constructed genome and annotation data. Morphological and transcriptomic analysis of ethylene seems to demonstrate it.

We will survey whether ABA (and auxin) regulate heterophylly in response to submergence and other environmental factors or not in future works.

Comment 2. I disagree with the authors' interpretation of the expression of polarity genes. Although they suggest that all polarity genes are reduced, their results in Fig 5F contradict this interpretation. While the expression of all the adaxial genes is indeed repressed, the expression of abaxial genes is not simply repressed. Instead, the level of KANADI3 (KAN3) is increased after submergence or ethylene treatment. This result is consistent with Kim et al's paper (PloS Genet, 2018) showing that KAN3, the major KAN gene expressed higher than KAN1 or KAN2. It is possible that KAN3 is also the major adaxial gene in *R. aquatica*. In that case, because of genetic redundancy and dominance of KAN3, changes in leaf polarity may also affect heterophylly of *R. aquatica*. This can be easily checked in the RNA seq data by directly comparing the absolute value of KANs.

Response 2. We appreciate the reviewer's comment on this point. Although 3 *KANADI* gene expressions in *Ranunculus trichophyllus* (*RtKANa*, *RtKANb* and *RtKANc*) were examined in the past study, *KAN3* homolog expression was not measured. These 3 *RtKANs* were homologs of *AtKAN1*, *AtKAN2*, and *AtKAN4* respectively.

Also, the function of *KAN3* on leaf development is not fully demonstrated. In *A. thaliana* (Eshed, Yuval et al. 2004), *KANs* single mutant showed no drastic change in polarity and *kan1 kan2* double mutant showed a defect in leaf polarity. *kan1 kan2 kan3* triple mutant seems similar phenotype. It suggests that *KAN1* and *KAN2* might mainly regulate leaf abaxial identity at least in *Arabidopsis* and also in related genus *Rorippa*.

We can't exclude or conclude the possibility that *KAN3* regulates leaf abaxial identity without functional analysis. However, because no obvious morphological change of polarity in submerged *Rorippa* leaves (Nakayama et al. 2014), we assumed that abaxialization as observed in *Ranunculus* is not involved in leaf form alternation in *R. aquatica*. On the other hand, expressional change of genes deciding adaxial and abaxial polarity might affect leaf form through regulation of cell proliferation and expansion.

We added a description of the up-regulation of one of *KAN1* and *KAN3s* and morphological observation of leaf polarity from the past study in discussion (Lines 401-403).

Responses to reviewer #3

Comment 1. As far as I could learn from literature, the species *Rorippa aquatica* is usually propagated clonally, even in nature. It would be good if you could add this information into the introduction since it is relevant. Did you get any hints from your genomic data and experiments whether the plant is able to produce seeds, and if not, what is the reason for it? It would be also good to have information on whether the chromosome pairs (in the diploid stage) are identical, i.e. homozygous as is the case for *Arabidopsis*, or whether they are heterozygous. If the species produces seeds, does this require different individuals, i.e. is it a non-selfing species?

Response 1. Thank you for your pointing out. We added a notation about seed production in this species (Lines 65-71). This species rarely produce seed in nature and in experimental condition. Although literature shows a possibility of self-incompatibility, the reason for infrequent seed production has not been tested; self-incompatibility or functional disorder of fertilization systems, this species-specific or common in the *Rorippa* genus.

About heterozygous or homozygous, because SNPs analysis with genome-seq data shows many heterozygous alleles, our *R. aquatica* clone is not a pure line or inbred strain. However, we only obtained cultured clone as aquarium plant and no data from natural populations, it is difficult to mention clearly about it in the main text.

Comment 2. In the introduction, I would like to suggest more references. For example, the second paragraph, lines 31 to 43, does not contain any reference, also the statement in lines 65-67.

Response 2. Thank you for your suggestion. We added some citations of references.

Comment 3. In discussion, it would be also good to include some facts on the conditions under water, i.e. low gas diffusion. This is the major factor that causes ethylene accumulation and therefore is the first signal for submergence in most plant species. This should be made clearer in the discussion, lines 352 to 357.

Response 3. Thank you for the important suggestion. We added a description of ethylene accumulation in response to submergence. It seems a critical factor to utilize as a submergence sensing signal.

Comment 4. Why did you not add *Nasturtium officinale* as a species into the comparisons? It should be even more related to *Rorippa* than *Barbarea* species. The genome sequence is not well annotated, but has been published here: Kiefer et al. 2019, Nature Plants.

Response 4. Thank you for providing important information. We checked the genome data of *Nasturtium officinale*. However, it does not suit comparative analyses with the *Rorippa* genus on the following points.

It is not closer to *Rorippa* than *Barbarea*. The report of Cai, L. and Ma, H. (2016) and our pre-analysis showed that *Nasturtium* and *Cardamine* were assigned the same phylogenetic cluster and different from clusters including *Rorippa* and *Barbarea*.

N. officinale might be polyploid. We analyze its genome using BUSCO, reporting that 52.9% of conserved single genes was duplicated in its genome.

We think that *B. vulgaris* is favored as the closest related species among (draft) genome available species and *C. hirsuta* is suitable for comparative analysis due to the availability of chromosome level genome.

Comment 5. Supplementary Figure 5, could you use two other colors for the *Rorippa* species? These two colors can be hardly distinguished from the *Eutrema* and *Arabidopsis* data.

Response 5. Thank you for the suggestion. We replaced it to improve visibility.

Comment 6. The link to the *Barbarea* genome data is no longer functional (line 559), please check and provide a correct link.

Response 6. Thank you for your pointing out. We modify it. Although we asked the corresponding author of *Barbarea* genome paper about the state of the genome database web page, we haven't gotten a response yet. We cited a public database in its genome assembly data submitted.

Comment 7. Please provide data on the species comparison, for example with a supplemental table listing all orthologs of all species, at least for the 10,845 genes that you used for further analyses (line 167). Only in this way other researchers can use your dataset.

Response 7. Thank you for the suggestion. We added ortholog analysis data as new Supplementary Dataset 1.

Comment 8. As far as I can see from the raw data IDs, part of the data have been used in another paper, presumably by the same group, namely the 1h and 4d time point, and the different light treatments after 1h. It is fine to use the data in another publication, but it should be clearly stated. Furthermore, it remains unclear which data have been added newly here, since the DRA IDs are not yet accessible. Please provide a Supplementary Table with read counts, mapping statistics and conditions for the RNAseq data used here, as you did in the other paper.

Response 8. We appreciate your pointing out. We added a list of read data submitted to the public database and their 1st appearance (this study or another paper) as new Supplementary Dataset 5 and it contains a summary of RNA-seq read mapping to *R. aquatica* genome.

Comment 9. Moreover, please also provide the full dataset. You identified more than 9,000 DEGs, but here you provide only the expression data for 143 up- and 82 down-regulated genes. At least, you should give the expression data (log₂FC, P-value, cpm) for the DEGs with ACC and AgNO₃ treatment.

Response 9. Thank you for the suggestion. We newly added whole expression profile data as Supplementary Dataset 3.

Comment 10. It is unclear how the treatment was actually done, and how long after treatment you harvested the tissue for RNAseq. The method part does not mention details in line 578. In line 435 you describe a treatment for two months, but this probably was used for making the pictures in Fig. 5. In addition, how did you apply the ACC, and how often? Did you spray it onto the plants? Did you spray only once? What was the composition of the spraying solution?

Response 10. Thank you for your pointing out. We fixed methods for chemical treatment. For all experiments, *R. aquatica* plants were kept in tanks continuously filled with water both under terrestrial and submerged condition with different water levels. For chemical treatment, water containing chemical were supplied to plants.

Comment 11. In line 232 and 585, you do not mention the cut-off that you used to define DEGs. Probably it is the same as in the previous paper.

Response 11. Thank you for your pointing out. We added a description about the DEG cutoff threshold to the results, method, and figure legends.

Comment 12. As far as I understood, you used also AgNO₃ treatment under submergence, but you do not present the data in Figure 5 or in the text. Why did you not comment on those data? Did they not fit into your hypothesis?

Response 12. Thank you for your comment. We added a description about the expression pattern under submergence with AgNO₃ (Lines 294-295). In the following analyses, we treated these 3 treatments (submerged 1 hour and 4 days and terrestrial ACC) as treatment inducing highly dissected (submerged type) leaves. Although AgNO₃ treatment inhibits this submergence response, it might not be completely reverse phenotypes. So it was excluded from this analysis for screening of shared genetic mechanism inducing submerged leaf.

Comment 13. Section on LMI1 and RCO (line 290 to 298), there is another RaRCO copy and two RaLMI1 copies. Do they behave similarly or differently?

Response 13. Thank you for your comment. *R. aquatica* gene name was defined based on similarity against *A. thaliana* protein. So, it has 4 *AtLMII* orthologs (RaChr03G09000 mentioned in main text, RaChr10G29540, RaChr10G29530 and RaChr11G26450). Former 2 genes show high protein identity (approx. 90%) to *AtLMII* and *Cardamine hirsuta LMII*. And Latter 2 gene show low identity to *AtLMII*(approx. 64%) and high identity to *ChisRCO* (85 and 90 %).

RaRCO homologs were significantly upregulated and *RaLMII* homologs were downregulated in response to submergence and one of *RaRCO* showed significant response for ACC treatment.

		Submergence	ACC
RCO homolog	RaChr03G09000 Referred in main text	Throughout upregulate	upregulate
	RaChr10G29530	Late upregulate	
LMI1 homolog	RaChr10G29540	Late downregulate	
	RaChr11G26450	Throughout downregulate	

Based on study from *Cardamine*, *RCO* orthologs in *R. aquatica* might be effective to leaf development than *LMII* homologs.

Comment 14. Figure S6 and section on blue light, is this the same dataset as in the previous paper, or a different one, for example a different time point? Please provide more details and give numbers of DEGs/ GOs and so on. A high percentage of genes that were shown in this figure do not meet the selection criteria of $\log_2FC > 1$ or < -1 . It is therefore doubtful whether the impact of blue light has such a big impact on this set of genes. So far, statistics are missing (see also next comment).

Response 14. Thank you for your comment. Source of RNA-seq data and Expression profiles were shown in new Supplementary Information 2 and 5. Genes shown in Fig. S6 are focused genes, not DEGs.

Comment 15. In Figure 5F and S6, you need to add statistics and standard deviations for the \log_2FC values. I assume that those data are from the RNAseq dataset and not from subsequent qRT PCR analyses? And the data from the AgNO₃ treatment are missing again.

Response 15. Thank you for your suggestion. As you mentioned, we added statistical data (significant difference $FDR < 0.01$). It shows focused gene data extracted from RNA-seq analysis. Because it is equivalent to heatmap data that mean \log_2FC and statistical significance were indicated by differences of color and character like “*” respectively in other papers, we think that showing these 2 values is sufficient.

Comment 16. Fig. 4A and 6A, can you provide quantitative data for your observations? With one picture from one leaf, one cannot conclude a lot. It should be easy to measure leaf thickness or area for Figure 6.

Response 16. Thank you for the important suggestion. About Fig.4, this observation was performed to estimate the time point for RNA-seq analysis (4 days after showing morphological change and 1 hour after as quick response). We think that statistics is not essential for this data.

About Fig 6A, the Dissection index was measured and used for statistical analysis. It indicates the complexity of the leaf form. Results of statistical analysis were added in the main text and Supplementary Figure 7.

Comment 17. Fig. 5B, the mock submergence treatment looks quite strange, compared with Fig. 1B. Can you add details to the legend about length of treatment?

Response 17. Thank you for your comments. It might be caused by differences in growth and physiological conditions. The leaf form of *R. aquatica* is influenced by various factors (temperature, light intensity, and quality and also the age of plants). So, different series of experiments are not comparable. However, we adjusted the experimental condition of each experiment series as much as possible to reduce differences.

Comment 18. Line 363 to 371 on GA, which genes are you referring to? Can you give a list or mention IDs? And you should mention the species names for which GA had no impact.

Response 18. Newly added Supplementary Dataset 2 includes data about the assignment of GO terms related to response to hormone stimuli. It seems too large number to denote in the main text.

We clearly denote species names showing different responses from *R. aquatica*.

Comment 19. Line 395 to 399, please provide more details on light quality changes underwater in nature. However, in your water depth, the changes in light quality might be minor.

Response 19. Thank you for your valuable comment. As you pointed out, the experimental submerged condition does not affect to light quality and quantity (reported in Ikematsu et al. 2023). We added it to main text (Lines 424-426).

Comment 20. There are some minor errors in the text, for example line 266, is the word "various" correctly used in this context; line 269, "one" should be "1"; line 316, remove "-".

Response 20. Thank you. We modified the text according to your suggestion.

Reviewers' comments:

Reviewer #1 (Remarks to the Author):

I have no further concern.

Reviewer #2 (Remarks to the Author):

The manuscript presents beautiful chromosomal painting data alongside comparative genomic analysis, leading me to conclude that *R. aquatica* originates from two *Rorippa* species through allotetraploidization. Unfortunately, the revised version has not satisfactorily addressed my concerns. It appears that the authors deliberately overlooked my comments, potentially influenced by their pre-existing model.

A notable issue lies in their treatment of transcriptome data. Despite clear evidence that ABA-responsive genes are downregulated immediately after submergence, the authors dismiss this finding. Instead, they emphasize the up-regulation of GA-responsive genes, even when some exhibit immediate down-regulation in their data. This selective interpretation compromises the objectivity of their conclusions, which is concerning.

Furthermore, the authors failed to consider a straightforward experiment to investigate whether ABA prevents leaf serration after submergence. This experiment, akin to ACC or AgNO₃ treatment, could provide additional valuable data, contributing to a more objective conclusion.

The revised manuscript continues to exhibit biased interpretations as I pointed out in the previous review. For example, while KAN1 and KAN3 are up-regulated in their data, the authors dismiss this as meaningless. Conversely, they interpret the down-regulation of HD-ZIPIIIIs as significant, aligning with their hypothesis. This inconsistency raises questions about the objectivity of their approach. It would not be surprising if HD-ZIPIIIIs are downregulated and KANADIs are upregulated after submergence since anyway, the submergence causes reduced leaf polarity. The authors should treat all data impartially rather than selectively validating their hypothesis.

In conclusion, I am inclined to recommend acceptance of the paper; however, I stress the necessity for a thorough revision to align the interpretation with the presented data.

Reviewer #3 (Remarks to the Author):

The revisions have considerably improved the manuscript. I still have some suggestions for improvements, as listed below.

Line 115 and line 449/ 454, you mention two different accessions used in this paper. However, only for one experiment it was specified which accession was used (chromosome painting). Can you clearly indicate in the method section, which accession was used for which experimental set-up? Especially important would be to indicate from which genotype the DNA and RNA came from that was used for the sequencing runs. I understood from the main text that accession N was used for DNA, was this also used for all RNA extractions?

Since you re-annotated the *Rorippa islandica* genome, and also used an annotation of the *Barbarea vulgaris* genome that is no longer available, it would be most helpful for other researchers to include the fasta sequences of the cds/ mRNAs that you have annotated and subsequently used to create Supplementary Dataset 1.

Results, first section, you describe here a second analysis of an already published dataset. In the earlier paper, you did SOM clusters, but basically have observed similar enriched GO terms. Please indicate in the text, for example in line 238, that this another analysis of the same RNAseq data, and include the citation of this paper here. Otherwise, some people might assume that this is an

independent dataset (although you mention the previous publication in Supplementary Dataset 5). For this reason, this section is rather a repetition of your earlier paper.

Some new text material would benefit from language editing. I do not understand sentence in line 229 or line 366. Line 333, word "while" could be removed. Line 423, "quite" is mis-spelled. Line 589, maybe some words missing – for example "The protein dataset consisted of 22 plant species ...". Line 625, re-phrase sentence a bit.

Figure 5, panel E, are you sure that you used the correct legend color code for up- and down-regulated genes? The up-regulated genes look rather red for 1 h submergence and ACC treatment which indicate down-regulation.

Dear reviewers

We highly appreciate you reviewing our manuscript (COMMSBIO-23-0769) entitled "Chromosome-level genome assembly of the model amphibious plant *Rorippa aquatica* reveals its allotetraploid origin and mechanisms of heterophyly upon submergence".

We took all the comments seriously and performed additional experiments during the revision. In accordance with the reviewer's comments, we revised the main text and added new data about ABA. Highlighted texts with pale blue in the revised manuscript indicate modified points in 2nd revision. The major changes are summarized below.

- We performed exogenous ABA treatment under submergence and added descriptions about the result with new Supplementary Fig. 7. We also added the explanation about GA.
- We modified descriptions about adaxial-abaxial polarity genes. We clearly denoted downregulation of adaxial *HD-ZIP III* genes and upregulation of some adaxial *KANADIs* and abaxialization based on these results. Discussion was also changed based on modified results.

Please find below our point-by-point responses to the reviewers' comments. We hope that our revised version addresses all of the concerns raised. We would, of course, be willing to make additional revisions to make the manuscript acceptable for publication.

Responses to Reviewer#2

Comment 1. A notable issue lies in their treatment of transcriptome data. Despite clear evidence that ABA-responsive genes are downregulated immediately after submergence, the authors dismiss this finding. Instead, they emphasize the up-regulation of GA-responsive genes, even when some exhibit immediate down-regulation in their data. This selective interpretation compromises the objectivity of their conclusions, which is concerning.

Furthermore, the authors failed to consider a straightforward experiment to investigate whether ABA prevents leaf serration after submergence. This experiment, akin to ACC or AgNO₃ treatment, could provide additional valuable data, contributing to a more objective conclusion.

Response 1.

Thank you for your suggestion. We apologize for misunderstanding your suggestions for GA responses in the previous round review.

According to your suggestions, we have checked the effect of ABA on the leaf shape of the submerged plants (Supplementary Fig. 7). The plants with ABA under submergence dyed within approximately 1 month. So, we can't produce results from the same condition as AgNO₃ treatment. Although it is a shorter treatment, we found that ABA inhibited submergence-responded leaf phenotype and suppressed leaf expansion, suggesting the involvement of ABA in regulating heterophylly. We added descriptions about ABA treatment (Lines 288-295, Lines 395-404, and Supplementary Fig. 7)

We apologize for misunderstanding about your suggestions for GA responses in previous round review. We didn't analyze GA in detail in this paper because our previous studies have already shown that GA is involved in the regulation of leaf morphology in *R. aquatica*. We added the discussion about GA since GO enrichment analysis showed GA response to submergence. At early response, GA-responsive genes were enriched in both up-(included throughout response) and down-regulated DEG at similar levels. So, it is difficult to show a clear increase or decrease at early response, although responsiveness is obvious. At late stage, only upregulated DEGs were enriched. So, we assumed increased GA signaling, at least in the late stage. We modified the discussion about GA response to submergence (Lines 409-415)

Comment 2. The revised manuscript continues to exhibit biased interpretations as I pointed out in the previous review. For example, while KAN1 and KAN3 are up-regulated in their data, the authors dismiss this as meaningless. Conversely, they interpret the down-regulation of HD-ZIPIIIIs as significant, aligning with their hypothesis. This inconsistency raises questions about the objectivity of their approach. It would not be surprising if HD-ZIPIIIIs are downregulated and KANADIs are upregulated after submergence since anyway, the submergence causes reduced leaf polarity. The authors should treat all data impartially rather than selectively validating their hypothesis.

Response 2. Sorry for our imprecise descriptions. We modified vague descriptions about the results of polarity-related gene and denoted clearly abaxialization from expression pattern of *HD-ZIP III* and *KANs* (Lines 326-332). Also in discussion, abaxialization in polarity genes and morphological effects were showed in parallel (Lines 424-431). Then we hypothesized about these difference (Lines 434-438)

Responses to reviewer#3

Comment 1. Line 115 and line 449/ 454, you mention two different accessions used in this paper. However, only for one experiment it was specified which accession was used (chromosome painting). Can you clearly indicate in the method section, which accession was used for which experimental set-up? Especially important would be to indicate from which genotype the DNA and RNA came from that was used for the sequencing runs. I understood from the main text that accession N was used for DNA, was this also used for all RNA extractions?

Response 1. Thank you for your pointing out. *R. aquatica* accession N were used for almost all experiments except for chromosome painting. I added descriptions about used accession in “Plant material” section of methods.

Comment 2. Since you re-annotated the *Rorippa islandica* genome, and also used an annotation of the *Barbarea vulgaris* genome that is no longer available, it would be most helpful for other researchers to include the fasta sequences of the cds/ mRNAs that you have annotated and subsequently used to create Supplementary Dataset 1.

Response 2. Thank you for your suggestion. We deposited results of *R. islandica* genome assembly and its annotation to Figshare (10.6084/m9.figshare.24849201). It will become in public and its address was added in Data availability section (Lines 664-665).

Comment 3. Results, first section, you describe here a second analysis of an already published dataset. In the earlier paper, you did SOM clusters, but basically have observed similar enriched GO terms. Please indicate in the text, for example in line 238, that this another analysis of the same RNAseq data, and include the citation of this paper here. Otherwise, some people might assume that this is an independent dataset (although you mention the previous publication in Supplementary Dataset 5). For this reason, this section is rather a repetition of your earlier paper.

Response 3. Thank you for your suggestion. We added description about use of RNA-seq read data used in previous published paper with citation (Lines 238-240). Although

partial RNA-seq read dataset was used in previous paper, other datasets were newly obtained, and screening method was different from that in previous paper. So, it is better to describe as a part of another analyses series rather than citation of results from previous paper.

Comment 4. Some new text material would benefit from language editing. I do not understand sentence in line 229 or line 366. Line 333, word "while" could be removed. Line 423, "quite" is mis-spelled. Line 589, maybe some words missing – for example "The protein dataset consisted of 22 plant species ..." Line 625, re-phrase sentence a bit.

Response 4. Sorry for our miswriting. We fixed the sentences according to your pointing out and checked texts thoroughly.

Comment 5. Figure 5, panel E, are you sure that you used the correct legend color code for up- and down-regulated genes? The up-regulated genes look rather red for 1 h submergence and ACC treatment which indicate down-regulation.

Response 5. Heatmaps in Fig. 5E visualized the scaled/normalized data to remove influence of too high difference between treated and control sample. Because scaling performed separately to up-regulated and down-regulated gene dataset, the value dose not correlate among these 2 panels.

REVIEWERS' COMMENTS:

Reviewer #2 (Remarks to the Author):

The authors revised the manuscript well and addressed all of my concerns. The paper would provide a useful aspect of how amphibious plant genomes have evolved. I am pleased to recommend the paper.

Reviewer #3 (Remarks to the Author):

The authors have adequately addressed my suggestions. However, the newly written sections in blue (and sometimes in yellow) still contain some grammatical and style errors (see my list below). Please correct the English style again, possibly with the help of a near-native speaker. This would strongly help the reader to understand the content that you would like to present.

Line 229, remove "following" or re-phrase.

Line 239, not easy to understand and "previous" is mis-spelled.

Supplementally Figure 7 -> mis-spelled, and in 4th line, modify as follows: "Plants were transferred from". See also Line 946 in the main document.

Supplementary Figure 9A -> why do you have two groups with CTR1? Are these two isoforms?

Line 295, I do not understand the phrase "due to the inhibition of the submergence response", could this be removed? I think that ABA generally reduces plant growth.

Line 328-329, re-phrase this sentence. Further in this Line, to whom do you refer by "They"? Probably the authors from ref. 6? Please clarify.

Line 342 -> shown rather than showed? This new section (until line 347) should be corrected in terms of English style.

Line 380, re-phrase the part "It have uncovered".

Line 390-393, the word "because" is not required here. These sentences should be re-phrased, while taking into account what is cause and what is consequence.

Line 395, please re-phrase.

Dear reviewers

We highly appreciate you reviewing our manuscript (COMMSBIO-23-0769) now entitled " A chromosome-level genome assembly for the amphibious plant *Rorippa aquatica* reveals its allotetraploid origin and mechanisms of heterophylly upon submergence ".

We took all the comments seriously and performed additional experiments during the revision. We revised the main text in accordance with the reviewer's comments. The major changes are summarized below.

- Manuscript title was changed to "A chromosome-level genome assembly for the amphibious plant *Rorippa aquatica* reveals its allotetraploid origin and mechanisms of heterophylly upon submergence".
- Text in manuscripts checked and modified thoroughly with supports by reviewers and English editing.
- Small modifications were performed according to paper's guideline.

Please find below our point-by-point responses to the reviewers' comments. We hope that our revised version addresses all of the concerns raised. We would, of course, be willing to make additional revisions to make the manuscript acceptable for publication.

Response to Reviewer #3

Comment 1. Line 229, remove "following" or re-phrase.

Response 1. Thank you for your pointing out. We modify the sentence according to your suggestion.

Comment 2. Line 239, not easy to understand and "previous" is mis-spelled.

Response 2. Thank you for the suggestion. We rewrote the sentence.

Comment 3. Supplementally Figure 7 -> mis-spelled, and in 4th line, modify as follows: "Plants were transferred from". See also Line 946 in the main document.

Response 3. Thank you for your pointing out. We modified the title and the sentence in the legend and main text about Supplementary Figure 7.

Comment 4. Supplementary Figure 9A -> why do you have two groups with CTR1? Are these two isoforms?

Response 4. Sorry for our carelessness. One of them is a misspelling of ETR1. We modified it.

Comment 5. Line 295, I do not understand the phrase "due to the inhibition of the submergence response", could this be removed? I think that ABA generally reduces plant growth.

Response 5. Thank you for suggestion. We removed the phrase "due to inhibition –" according to your suggestion.

Comment 6. Line 328-329, re-phrase this sentence. Further in this Line, to whom do you refer by "They"? Probably the authors from ref. 6? Please clarify.

Response 6. Sorry for vogue text. It means expression profiles of HD-ZIP IIIs and KANs. We modified it.

Comment 7. Line 342 -> shown rather than showed? This new section (until line 347) should be corrected in terms of English style.

Response 7. Thank you for suggestion. We modified this sentence according to suggestion and reconstructed this section (Lines 341-348).

Comment 8. Line 380, re-phrase the part "It have uncovered".

Response 8. Thank you for your pointing out. We modified the sentence (Lines 381-382).

Comment 9. Line 390-393, the word "because" is not required here. These sentences should be re-phrased, while taking into account what is cause and what is consequence.

Response 9. Thank you for your suggestion. We modified this part (Lines 391-396).

Comment 10. Line 395, please re-phrase.

Response 10. Thank you for your suggestion. We modified this sentence (Line 396)